

# Offshore wind farms modify low-level jets

Daphne Quint[1,2], Julie K. Lundquist[1,2,3,4], and David Rosencrans[1]

[1]Department of Atmospheric and Oceanic Sciences, University of Colorado Boulder, Boulder, Colorado, 80309-0311, United States
[2]National Renewable Energy Laboratory, Golden, Colorado, 80401, United States
[3]Renewable and Sustainable Energy Institute, Boulder, Colorado, 80309, United States
[4]Johns Hopkins University, Baltimore, Maryland, 21218, United States

**Correspondence:** Julie K. Lundquist (julie.lundquist@colorado.edu)

**Abstract.** Offshore wind farms are scheduled to be constructed along the east coast of the United States in the coming years. Low-level jets (LLJs) – layers of relatively fast winds at low altitudes – also occur frequently in this region. Because LLJs provide considerable wind resources, it is important to understand how LLJs might change with turbine construction. LLJs also influence moisture and pollution transport; thus, the effects of wind farms on LLJs could also affect the region's meteorology.
We compare one year of simulations from the Weather Research and Forecasting model with and without wind farms incorporated, focusing on locations chosen by their proximity to future wind development areas. We develop, present, and validate an algorithm to detect LLJs at each hour of the year at each of these locations. Offshore LLJs in this region occur most frequently at night, in the spring and summer months, in stably stratified conditions, and when a southwesterly wind is blowing. LLJ wind speed maxima range from 10 m s$^{-1}$ to over 40 m s$^{-1}$. The altitude of maximum wind speed, or jet "nose" predicted by WRF,
is typically 300 m above the surface, above the height of most profiling lidars. Wind farms erode LLJs, as fewer LLJs occur in the wind farm simulations than in the no-wind-farm (NWF) simulation. When LLJs do occur in the simulation with wind farms, their noses are higher than in the NWF simulation: the LLJ nose has a mean altitude near 300 m for the NWF jets, but that nose height moves higher in the presence of wind farms, to a mean altitude near 400 m. Rotor region (30–250 m) wind veer is reduced across almost all months of the year in the wind farm simulations, while rotor region wind shear is similar in
both simulations.

## 1   Introduction

Wind farms off of the coast of the northeastern U.S. are expected to grow rapidly in the next few years, with goals to reach a capacity of 30 GW by 2030, and 110 GW by 2050 (U.S. Department of Energy, 2023). Wind turbines will be grouped into clusters within the 27 active wind farm lease areas that span the mid-Atlantic Outer Continental Shelf.

The considerable wind resource in this region derives not only from the faster wind speeds over open water but also from the frequent occurrences of low-level jets (LLJs) here. An LLJ is a region of relatively fast winds at low altitudes in the atmosphere. LLJs occur globally (Rife et al., 2010) and provide considerable wind resource due to their fast wind speeds (Vanderwende et al., 2015; Doosttalab et al., 2020). However, LLJs can also cause excess wear and tear on wind turbine blades (Kelley et al.,



2006; Gutierrez et al., 2016). The fast wind speeds, wind shear, and wind veer associated with LLJs can affect the energy
production of a wind farm (Chatterjee et al., 2022), especially when LLJs interact with complex terrain (Radünz et al., 2022).

Several mechanisms can induce the formation of LLJs. LLJ formation was first described for the onshore environment.
As described by Blackadar (1957) and Bonner (1968), LLJs are initiated by the evening cessation of turbulence near the
surface. This release of turbulent stresses or frictional decoupling leads to an acceleration, taking the form of an inertial
oscillation, which in turn results in a jet structure in low-level winds. Baroclinicity, due to sloping terrain or frontal dynamics
can supplement the inertial oscillation mechanism. Several climatologies of onshore LLJs explore the roles of these various
mechanisms and resulting impacts on jet characteristics such as the maximum wind speed, the height of the wind speed
maximum, and the evolution of the LLJ over the course of a night (Bonner, 1968; Whiteman et al., 1997; Banta et al., 2002;
Song et al., 2005; Lundquist and Mirocha, 2008; Rife et al., 2010; Baas et al., 2009; Vanderwende et al., 2015; Smith et al.,
2018, 2019). LLJs frequently cause strong wind shear, and in some cases wind veer, underneath the nose of the jet (Banta et al.,
2002).

The same mechanisms (inertial oscillations and baroclinicity) result in LLJs in the offshore environment, even as winds are
generally faster in the offshore environment due to the low surface roughness of the ocean. As first described in Högström and
Smedman-Högström (1984), flow from land over water also experiences a frictional decoupling, leading to an acceleration and
a jet structure in winds off a coast. The resulting marine LLJs were explored in more detail by Smedman et al. (1995) and
Smedman et al. (1993). The frequent occurrences of LLJs in the North Sea (Dörenkämper et al., 2015; Wagner et al., 2019)
arise from this mechanism. LLJs can also form in coastal regions after the cessation of sea breeze circulations, triggered by an
inertial oscillation initiated when the sea breeze decays (Angevine et al., 2006). As in the onshore case, baroclinicity, in this
case due to temperature contrasts between land and water in coastal regions, can amplify the LLJ. As a result, winds that allow
long offshore fetch – southwesterly winds in the U. S. east coast wind energy regions – are associated with LLJs in the New
York Bight region (Colle and Novak, 2010). Southwesterly winds occur frequently in this region, and arise from geostrophic
flow around persistent high pressure over the ocean and low pressure over land. This southwesterly flow brings relatively
warm air from the south over colder mid-Atlantic waters, creating stable conditions suitable for frictional decoupling and jet
formation. At night during the summer, radiative cooling on sloping terrain to the west creates an air temperature gradient
near the surface, with higher temperatures over the sea. This baroclinicity impacts the gradient of the geostrophic wind in
the vertical, resulting in a thermal wind that increases with height. The LLJ is formed at the local maximum in wind speeds
associated with this thermal wind gradient (Holton, 1967; de Jong et al., 2023). Additionally, low-pressure systems moving
over land to the west can result in a tightening of the pressure gradient and a stronger southwesterly wind offshore (Strobach
et al., 2018).

Previous studies have characterized offshore LLJs in this region from observations, albeit for case studies during specific
field campaigns. Using low-level slant soundings from the Long-EZ aircraft over two days south of Martha's Vineyard, Mahrt
et al. (2014) observe low-level wind maxima associated with developing stable stratification; the altitude of the wind speed
maximum is higher with stronger stability. Farther north, in the Gulf of Maine, Pichugina et al. (2017) used a ship-based





scanning lidar to assess LLJ profiles during an intensive one-month deployment, finding that LLJs occurred 63 % of the time, with maximum jet heights occurring up to 600 m, although the average jet height was approximately 155 m above the surface.

More recently, offshore floating lidar buoys have enabled longer-term characterization of LLJs, but at altitudes constrained by lidar capabilities. In an analysis of LLJs at two floating lidar buoys, Debnath et al. (2021) find that LLJs occur primarily when a southwesterly wind is blowing, in the spring and summer months, and when there is a positive air–sea temperature difference. Similarly, Aird et al. (2022) identify LLJs in several groups along the east coast and finds LLJs are preferentially aligned with the nearest coastline, occur primarily in the warm season with a peak in June, and when air temperatures are

cooler offshore. Both find that LLJs often occur at heights relevant to wind energy. Floating lidar buoy studies are limited to the lowest 220 m accessible by lidar measurements.

The interaction of LLJs with wind farms in the offshore U. S. east coast regions has not yet been investigated, despite the occurrences of LLJs in this region and the planned offshore development. In addition to providing wind resources, LLJs can affect the recovery of wind farm wakes, which result from clusters of turbines. To make maximum use of the wind lease

areas, wind turbines are arranged in large clusters that often interact with each other. Large arrays of turbines produce wakes – regions of slower wind and more turbulent flow downwind of the turbines – that can propagate for tens of kilometers (Hasager et al., 2006; Platis et al., 2018; Rosencrans et al., 2023). The wake wind speed reduction downwind of the wind farm can reduce power output (Nygaard, 2014; Lundquist et al., 2019). Wakes recover more slowly during stably stratified conditions (Lundquist et al., 2019; Cañadillas et al., 2022). Wakes also recover faster in the presence of a low-height LLJ profile, which

improves the performance of turbines downwind (Gadde and Stevens, 2021). A simulation study of the interactions of one LLJ with wind farms in the North Sea (Larsén and Fischereit, 2021) suggests that the position of the nose of the LLJ and the resulting shear between the surface and the LLJ nose are modified by the wind farms and their wakes.

This research broadens that understanding by using a yearlong data set of numerical weather prediction simulations to quantify the expected impact of wind farms on LLJs offshore. We assess the temporal and spatial variability of occurrences of

LLJs and the characteristics of LLJs such as their maximum wind speed, height of wind speed maximum, and associated wind shear and wind veer. We then assess how wind turbine wakes influence these LLJ characteristics. Here, however, we compare characteristics of LLJs both with and without wind farms present based on numerical simulations, which enable assessment of LLJ structures that occur above the height of profiling lidars. Understanding the interaction between wind farms and LLJs is crucial for accurately forecasting energy production.

To understand how wind turbines may modify LLJs, we compare the Weather Research and Forecasting (WRF) model (Skamarock et al., 2021) simulations of Rosencrans et al. (2023) that represent conditions in the region from September 2019 to September 2020. Three of these simulations include different wind farm layouts, and one simulation does not include any wind farms in the model. In Sect. 2, we present the data set used and the locations of interest. In Sect. 3, we describe our definition of a low-level jet and our detection methods. Section 4 includes quantification of LLJ occurrences, seasonal and

diurnal cycles, jet heights, rotor region wind veer and wind shear, and wind direction for both the no-wind-farm and wind farm simulations. We discuss an extreme LLJ case study in Sect. 5. We conclude and summarize our results in Sect. 6. The results of this assessment are neither intended to make nor suitable for making commercial judgements about specific wind projects.





## 2 Data

### 2.1 Model data set

In this study, we use the NOW-WAKES data set described in Rosencrans et al. (2023) and Bodini et al. (2023), which quantified the effects of wind farm wakes on energy production. Full details on the data set can be found in Rosencrans et al. (2023), but a brief summary is provided here. The data set was created using WRF version 4.2.1, and the wind farm parameterization of Fitch et al. (2012). We use the innermost domain (domain 2) of the two nested domains, which is bounded by 76.208° W–64.977° W and 37.389° N–42.137° N (Fig. 1). Domain 2 has a 2 km horizontal resolution and a 10 m vertical resolution near the surface

with stretching aloft. There were 34 vertical levels in the lowest 2000 m, and 29 in the lowest 750 m. These simulations without wind farms have been validated in comparison to floating lidar observations at two locations in the domain (Rosencrans et al., 2023). The period from 1 September 2019 00:00 UTC to 31 August 2020 23:50 UTC provides a temporal resolution of 10 minutes; we used hourly time steps for our analysis.

Three simulations consider three different wind farm layouts, and one simulation does not include any wind farms in the

model (Table 1). The first wind farm layout includes only turbines in one specific lease area, referred to as "ONE" here and in Rosencrans et al. (2023). The second layout considers turbines in all lease areas, and the third considers all turbines in both the call areas and the lease areas as of September 2019. Figure 1 shows the locations of the wind turbines for each of these simulations. The turbines introduced in these simulations are 12 MW with a rotor disk extending from 30 m to 245 m and a hub height of 138 m. To assess the effect of including or not including turbine-generated turbulent kinetic energy (TKE),

separate simulations with 0 % and 100 % added TKE are available, but we consider only the 100 % TKE simulations. Note that the simulation for the call areas only lasts 4 months (September and October of 2019, and July and August of 2020) instead of the full year in the other simulations due to computational limitations.

**Table 1.** Summary of each simulation

| Simulation Type | Acronym | Turbine Type | Time | Added TKE | # Turbines |
|---|---|---|---|---|---|
| No wind farms | NWF | N/A | 09/2019 - 09/2020 | N/A | 0 |
| ONE only | ONE100 | 12 MW | 09/2019–09/2020 | 0 % and **100 %** | 177 |
| Lease Areas | LA100 | 12 MW | 09/2019–09/2020 | 0 % and **100 %** | 1418 |
| Call Areas | CA100 | 12 MW | 09/2019–11/2019, 07/2020–09/2020 | **100 %** | 3219 |

### 2.2 Locations of detailed study

We focused on five locations in the vicinity of the ONE area, call areas, and lease areas (Fig. 1). Two of these locations are

discussed in detail (ONE centroid (ONEcent) and southern lease area centroid (southLA)), while figures for the other three





locations (SW corner of ONE (SWcorner), NE buoy, and SW buoy) can be found in the appendix. We chose to focus on the ONEcent and the southLA centroid because they represent the two geographic extremes of the data set. Differences between the no wind farm (NWF) and the wind farm (WF) simulations diminished with distance from the wind farm, so locations on land are not addressed here. The ONEcent and SWcorner locations were analyzed using the ONE-only simulation (ONE100). The locations of two floating lidar buoys (NE buoy and SW buoy) are analyzed using the call area simulation (CA100). The southLA centroid is analyzed using the simulation including all lease areas (LA100).

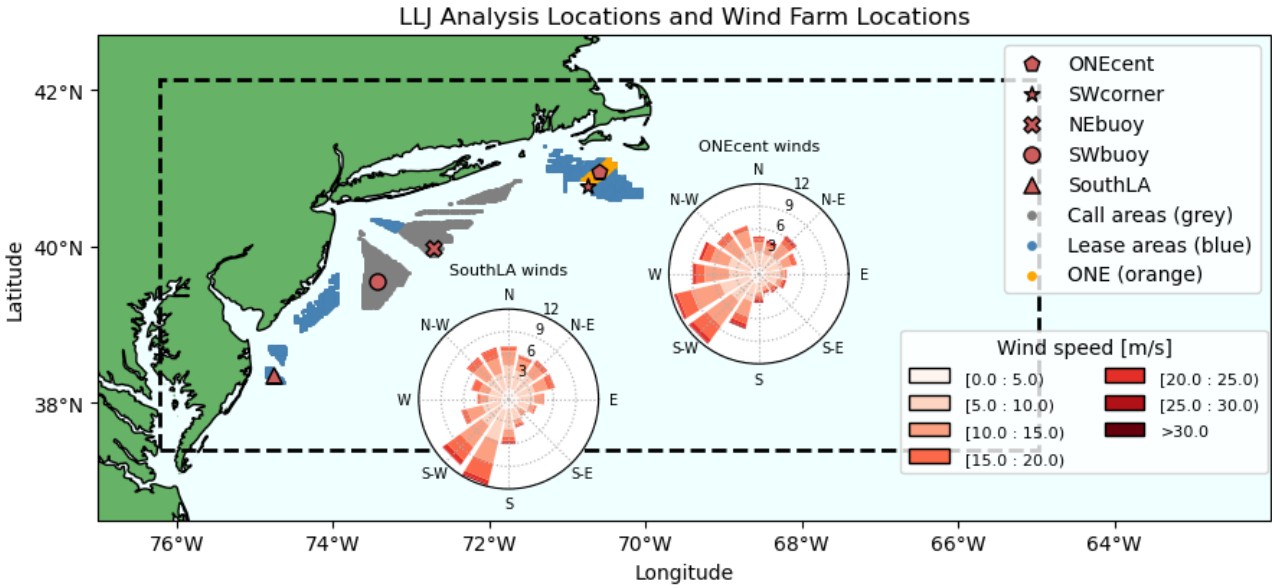

**Figure 1.** Simulation domain including turbine locations and regional wind roses. The five locations analyzed are in red with a black outline. The turbine locations for each of the three wind farm layouts are shaded, with ONE in orange, Lease Areas in blue, and Call Areas in gray. The domain 2 boundary is outlined with the dashed black line. The wind roses show the wind speed (shading) and direction (angle) at 130 m for 1 year at ONEcent (top right), and at southLA (bottom left). The distance from the center of the plot indicates the percentage of values in each bin.

**Table 2.** Coordinates for each location on the map, where LA stands for Lease Area. Locations in bold are the focus of this paper.

|           | **ONEcent** | SWcorner  | NEbuoy    | SWbuoy    | **SouthLA** |
|-----------|-------------|-----------|-----------|-----------|-------------|
| Latitude  | 40.95° N    | 40.77° N  | 39.97° N  | 39.55° N  | 38.35° N    |
| Longitude | 70.59° W    | 70.74° W  | 72.72° W  | 73.43° W  | 74.76° W    |





## 2.3 Additional locations

To assess the geographic variability of LLJs, LLJs were also assessed at each hour from 1 September 2019 to 31 August 2020 at each of the points in Fig. 2 using the same NOW-WAKES data set as described above. Analysis of LLJs at these locations
quantifies the spatial variability in the region. The 126 points are spaced approximately 26 km apart, excepting any points on land.

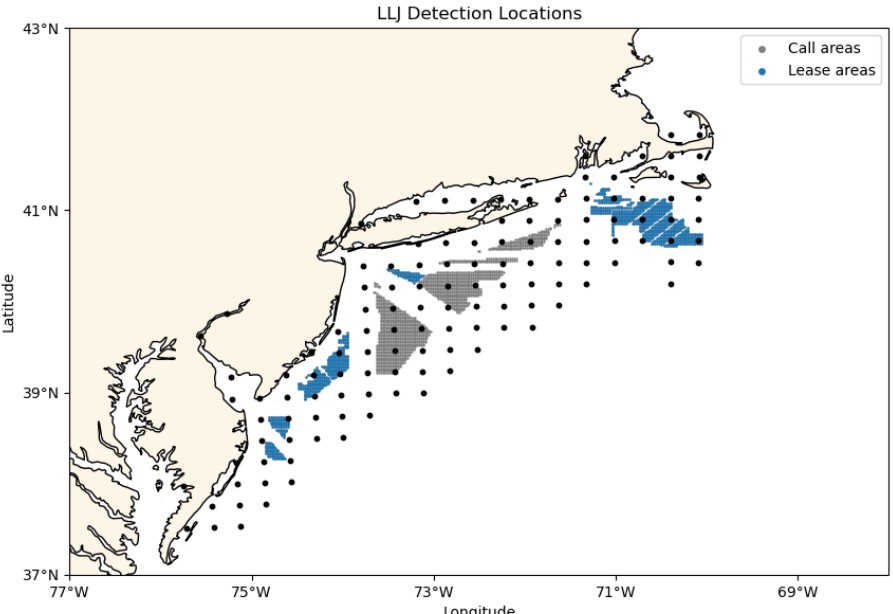

**Figure 2.** Locations where LLJs were assessed are marked by the black points. The wind lease areas and the call areas are marked in blue and gray, respectively.

## 3 Methods

### 3.1 LLJ detection

To identify LLJs in the simulations described above, we follow the established methodology described by Vanderwende et al.
(2015), based on Whiteman et al. (1997) with slight modifications for the offshore environment. LLJs are detected if the maximum wind speed occurs in the lowest 750 m of the atmosphere and is at least 10 m s$^{-1}$. The wind speed reduction above this wind speed maximum (the "nose" of the jet (Banta et al., 2002)) must be at least 3 m s$^{-1}$; we considered heights up to 2 km for our analysis. Given the difference in mechanisms offshore and onshore (smaller force of friction leading to weaker super-geostrophic acceleration), we use a smaller shear threshold than in Vanderwende et al. (2015).
Further, given the relevance of LLJs occurring within the rotor layer (Gadde and Stevens, 2021), we also define very low-level jets (vLLJs) that in addition to the above criteria, must have a jet nose height below 260 m. vLLJs occur at heights more





relevant to wind energy than LLJs higher in the atmosphere, and, as shown below, their interactions with wakes are different from LLJs with nose heights at higher altitudes.

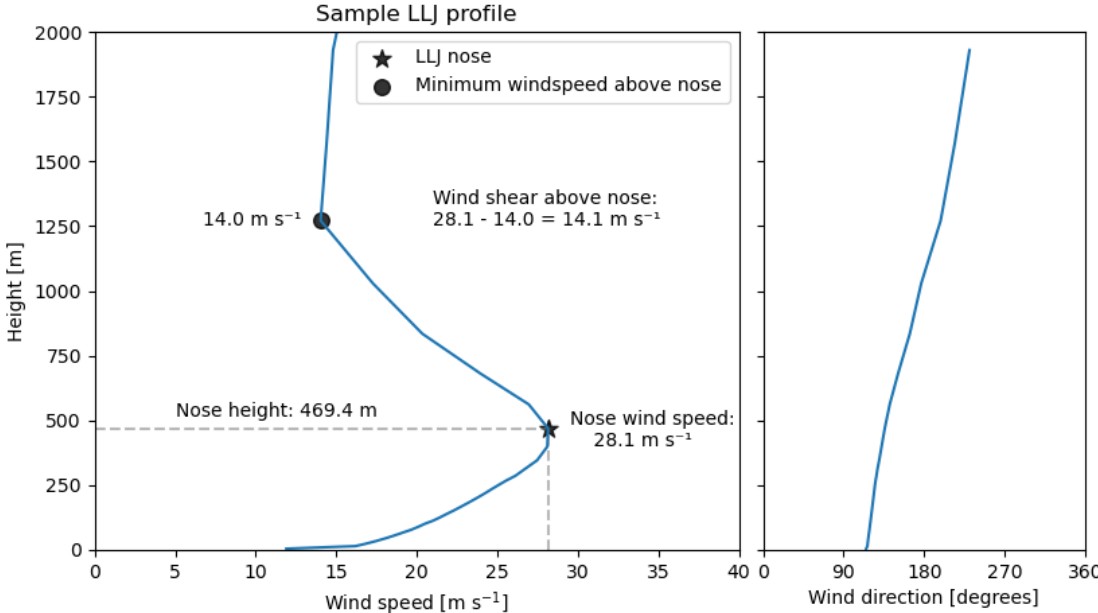

**Figure 3.** A sample LLJ profile. Labeled on the plot is the nose height, nose wind speed, and wind shear above the nose. The secondary plot on the right shows a profile of the wind direction with height. This sample is from 31 December 2019 at 0500 UTC in the no-wind-farm simulation at ONEcent.

### 3.2 LLJ classification

When an LLJ is identified, it is given a classification from LLJ-0 to LLJ-3 based on the wind speed and wind shear values, as described in Table 3 and as seen in an LLJ-3 profile (Fig. 3). This classification pattern follows that of Vanderwende et al. (2015) and Whiteman et al. (1997). These values best define the magnitude of the LLJ, and help to determine the potential impact on wind energy.

### 3.3 Stability calculation

LLJs onshore tend to occur in stably stratified conditions. To look for a similar relationship here, we calculate the Obukhov length ($L$):

$$L = -\frac{u_*^3 \overline{\theta_v}}{\kappa g (\overline{w'\theta_v'})} \tag{1}$$





**Table 3.** Low-level jet classifications were determined using the scheme in the table.

|       | Minimum Wind Speed [m s$^{-1}$] | Above Nose Shear [m s$^{-1}$] |
|-------|--------------------------------|-------------------------------|
| LLJ-0 | 10                             | 3                             |
| LLJ-1 | 12                             | 5                             |
| LLJ-2 | 16                             | 8                             |
| LLJ-3 | 20                             | 10                            |

where $u_*$ is the friction velocity, $\theta_v$ is the virtual potential temperature, $\kappa$ is the von Kármán constant of 0.4, $g$ is gravitational acceleration, and $\overline{w'\theta'_v}$ is the vertical turbulent surface heat flux. Values between 0 m and 1000 m are considered stably stratified conditions, and values from -1000 m to 0 m are considered unstable. Values outside of this range are considered neutral (Muñoz-Esparza et al., 2012).

Although $L^{-1}$ (referred to as RMOL) is an available output from the WRF simulations, we instead calculate $L$ using Eq. (1) and WRF output values of UST (friction velocity) and HFX (surface heat flux). The value of RMOL output by WRF is calculated in the surface layer scheme, whereas the value of the heat flux is calculated afterward in the land surface module, and so may be different from the value used in the surface layer scheme (Olson, 2023). The planetary boundary layer scheme, which induces the winds, responds to the value of the heat flux output by the land surface module, so the value of $L^{-1}$ calculated with Eq. (1) is more relevant to the winds.

RMOL and $L^{-1}$ can differ. When comparing the calculated values of $L$ to the WRF output value of RMOL at ONEcent over a year of the NWF simulations, we found that the stability classifications (stable, unstable, or neutral) disagree 5.66 % of the time at ONEcent (Fig. 4).





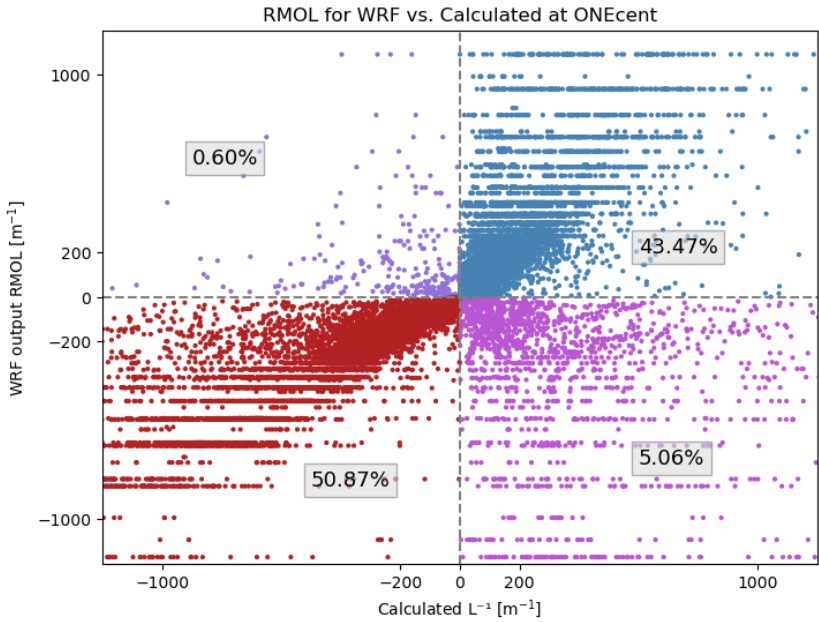

**Figure 4.** Comparison of $L^{-1}$ calculated by Eq. 1 from WRF output UST and HFX (x-axis) and the WRF output variable RMOL (y-axis) at ONEcent. The percentage of points that fall into each quadrant (divided by the dashed gray lines) are represented by the text in each quadrant.

## 3.4 Data set validation for LLJs

To assess the skill of WRF in simulating LLJs, we treat this case as a dichotomous forecast: an LLJ either occurred or it did not. We compare WRF simulation results to observations at the E06 floating lidar, which is located at 73.43° W 39.55° N. The profiling lidar includes horizontal wind speed and direction data for 10 vertical levels from 18m to 198m, with an interval of

20m. For each of the 8516 hours of the year that the lidar was operational, we applied a modified LLJ detection algorithm to both data sets and compared the frequency and timing of the occurrences of LLJs. Given the shallow layer of observational data available and the fact that many LLJs occur above the highest lidar level, we reduce the wind speed reduction above the nose threshold from 3 m s$^{-1}$ to 1 m s$^{-1}$. LLJs are relatively rare in this layer of the atmosphere: of the 8516 hours analyzed, 145 events were identified by the lidar, and 109 were identified by WRF.

We first follow the validation used in Aird et al. (2022), which compares the mean LLJ profile and the monthly distribution of LLJs for both datasets (Fig. 5). Mean LLJ wind speeds and standard deviations for each level are similar between WRF and lidar above 100m. Below this, WRF tends to have faster wind speeds, with a maximum difference of around 2 m s$^{-1}$ near the surface. Standard deviations are also slightly larger near the surface in the WRF data set. Both data sets show a mean nose height of just over 100m and and mean nose wind speed of around 13 m s$^{-1}$. Both data sets also have a similar seasonal cycle,

with peaks in June, and low occurrences from August to February. The largest differences between both data sets occur from January to April, with April having around a 5% difference in LLJ occurrences.





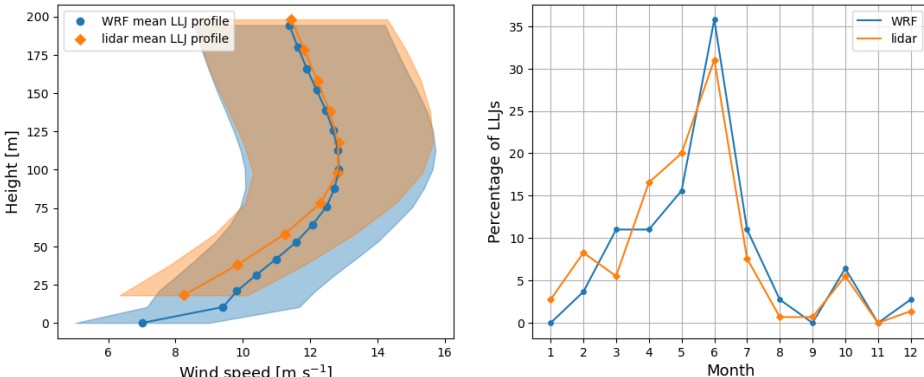

**Figure 5.** Left: Mean LLJ profile for WRF (blue circle) and lidar (orange diamond), with shading for the 1 standard deviation range. LLJs were both calculated at the E06 floating lidar during the same time period.

We also assess how the timing of LLJs compares between WRF and lidar by considering each individual hour of the year. We determine the accuracy, bias, probability of detection, false alarm ratio, probability of false detection, success ratio, and threat score using LLJ data from September 2019 - September 2020 (Table 4). Metrics in the last 7 rows of the table are calculated
using the hits, correct negative, misses, and false alarm metrics defined in the previous 4 rows. Because of the rarity of these events, the accuracy and probability of false detection scores can be misleading due to the high number of true negative results identified in both the model and observations. WRF has fewer events overall, corresponding to a bias less than 1. WRF tends to have higher LLJ nose heights which may not be detectable by the lidar. This could lead to fewer events detected overall in the WRF simulations. WRF performs poorly in the probability of detection, false alarm ratio, success ratio, and threat score
metrics due to the high occurrence of misses and false alarms. While WRF profiles often match lidar profiles well, there are times when WRF misses. In the left panel of Figure 6, the WRF profiles matches the lidar profile well. In the right panel, WRF does not simulate conditions below 200m well. The center panel demonstrates a case where the WRF has a similar profile to the lidar, but has a nose height that is too high. Within the lowest 200m of the atmosphere WRF does not have an LLJ due to insufficient shear above the nose. When we instead consider a 400m layer, a LLJ profile can clearly be identified.

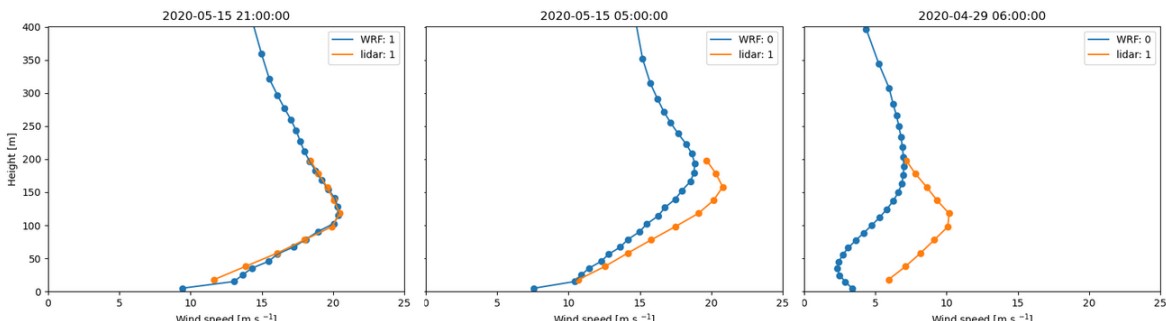

**Figure 6.** Wind profiles for WRF (blue) and lidar (orange) on 15 May 2020 at 21:00 UTC (left), 15 May 2020 at 05:00 UTC (center), and 29 April 2020 at 06:00 UTC (right). Wind speed is on the x-axis, and height above the surface in on the y-axis. A zero in the legend indicates that a LLJ was not identified in the lowest 200m of the atmosphere, and a 1 indicates that an LLJ was identified. All profiles are at the E06 floating lidar buoy at 73.43° W, 39.55° N.

WRF does not closely match the lidar observations when considering each hour individually, as we identified many misses and false alarms. The data set used here follows best practices for simulating LLJs using WRF. Previous studies recommend a 2km resolution in the inner domain, a large number of vertical levels in the boundary layer, ERA5 reanalysis data for initial and boundary conditions, and use of the MYNN PBL scheme (Nunalee and Basu, 2014; Wagner et al., 2019; Kalverla et al., 2019; Siedersleben et al., 2018; Tay et al., 2021). Other studies have also used WRF to simulate LLJ events and also found variable agreement between WRF simulations and lidar observations (Colle and Novak, 2010; Vanderwende et al., 2015; Larsén and Fischereit, 2021; Aird et al., 2022).

It is difficult to validate the performance of WRF due to limited observational data. The lidar can only provide validation for heights up to 200m, but LLJs often occur higher than this (Aird et al., 2022; Vanderwende et al., 2015; Zhang et al., 2006). Our study aims to explore the variability of LLJs at heights relevant to wind energy, and WRF is currently the best option given the limitations of profiling lidars. Results in Figure 5 show that WRF performs reasonably well for characterizing LLJs on larger time scales.

# 4 Results

## 4.1 LLJ occurrences

In the NWF simulation, LLJs occur about 25 % of the time at each of the five locations. The ONEcent location had the most LLJs, and the SWbuoy had the fewest occurrences. Level 0 and 1 jets occur most frequently, while the faster level 2 and 3 jets are rarer (Fig. 7).

At each of the five locations, LLJs occur less frequently when wind farms are present (Fig. 7). The southLA location has the largest reduction in LLJs, with a 23 % overall decrease (so that LLJs occur only 19.5 % of the time), followed closely by the ONEcent location with a 21 % overall reduction (so that LLJs occur only 20.3 % of the time). LLJs at the SWcorner site are





**Table 4.** Summary of LLJ validation results

| Metric | Result | Interpretation |
|---|---|---|
| WRF # events | 109 | N/A |
| Lidar # events | 145 | N/A |
| Hits | 43 | There were 43 times when a LLJ was both forecast and observed |
| Correct negatives | 8305 | There were 8305 times when an LLJ was not forecast and not observed |
| Misses | 102 | There were 102 times when a LLJ was not forecast but an LLJ was observed |
| False alarms | 66 | There were 66 times when a LLJ was forecast but not observed |
| Accuracy | 0.9803 | Overall, 98.03% of forecasts were correct |
| Bias | 0.7517 | The forecast frequency of LLJs is less than the observed frequency of LLJs |
| Probability of detection | 0.2966 | 29.66% of observed LLJs were correctly forecast |
| False alarm ratio | 0.6055 | 60.55% of predicted LLJs were false alarms |
| Probability of false detection | 0.0079 | 0.79% of no-LLJ events were incorrectly forecast as LLJs |
| Success ratio | 0.3945 | 39.45% of forecast LLJs were actually observed |
| Threat score | 0.2038 | Forecast LLJs do not correspond to observed LLJs very well |

210    reduced significantly less in the WF simulation, with 10.7 % fewer LLJs in the WF simulation than in the NWF simulation. The disparity between the ONEcent and SWcorner sites likely occurs because there are few turbines upwind of the SWcorner site when southwesterly winds are blowing. When wind farms are present, level 3 jets are reduced less than for level 0, 1, and 2 jets at the ONEcent and southLA sites, with reductions of 6.45 % and 9.84 %, respectively (Table 5).



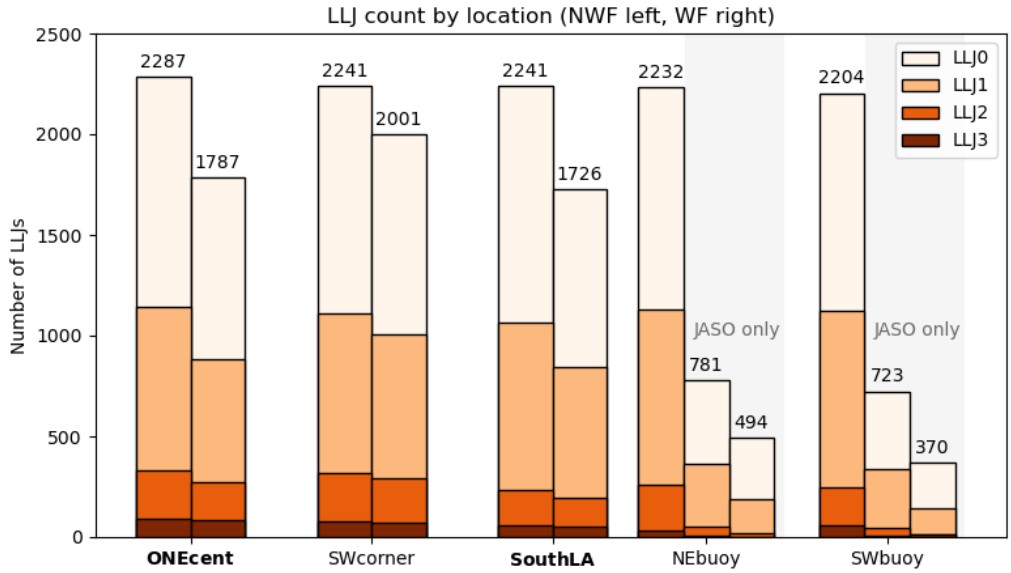

**Figure 7.** LLJ occurrences at each of the analyzed locations out of the 8784 possible hours. The text on top of each bar refers to the total number of hours that LLJs occurred at that location. The no-wind-farm counts are on the left, and the wind farm counts are on the right. LLJ category is shaded. Note that data from the NE buoy and the SE buoy include only July, August, September, and October (JASO) for the wind farm simulation, while the other three locations are for one full year. At the NE and SW buoys, the full year for the NWF simulation is also included on the far left. The right two bars are for the NWF and WF for JASO only.

**Table 5.** Percent reduction in LLJs between the NWF and WF simulation for each classification at three locations with a full year of data.

|          | ONEcent | SWcorner | SouthLA |
|----------|---------|----------|---------|
| LLJ0     | 21.2    | 12.2     | 25.1    |
| LLJ1     | 24.4    | 9.2      | 21.3    |
| LLJ2     | 22.1    | 8.79     | 21.0    |
| LLJ3     | 6.45    | 10.0     | 9.84    |
| All LLJs | 21.9    | 10.7     | 23.0    |

We find similar results for the array of LLJ points across the region (Figs. 8, 9, 10). In the NWF simulation, LLJs occur between 14 and 27 % of the time, with most locations experiencing LLJs around 25 % of the time. Locations far from any wind turbines see similar rates of LLJ occurrences in both the NWF and the LA100 simulations. Points in the vicinity of wind farms see up to a 30.5 % reduction in LLJ occurrences in the lease area simulation, and up to 49.2 % reduction in the call area simulation.

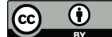

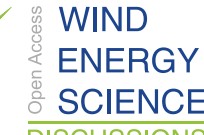

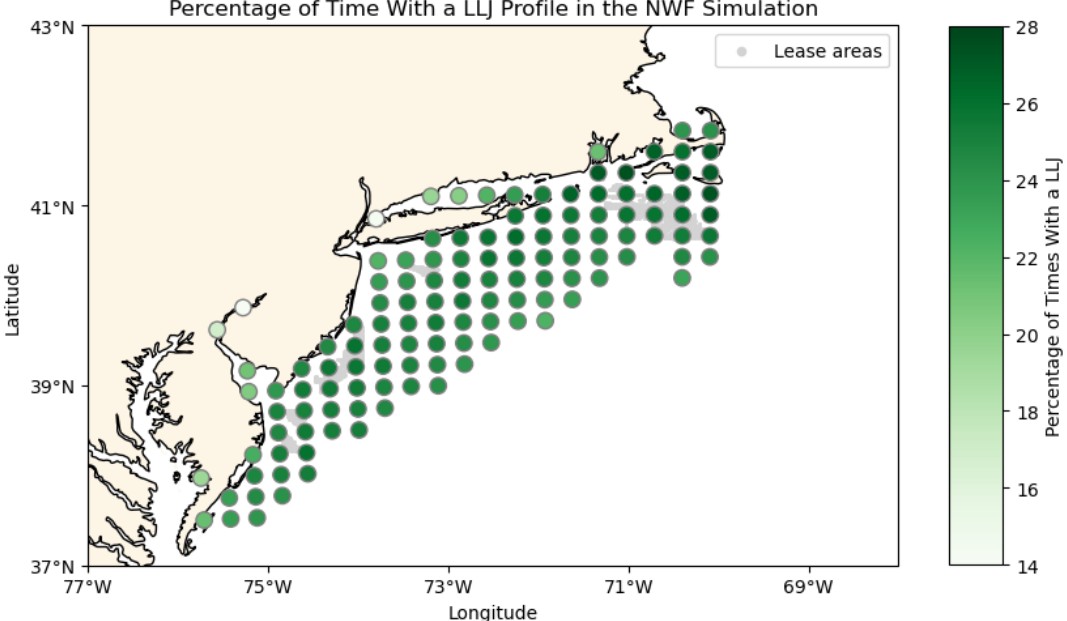

**Figure 8.** The percentage of times with an LLJ profile in the NWF simulation is shaded in green. Wind lease areas are shaded in gray in the background.



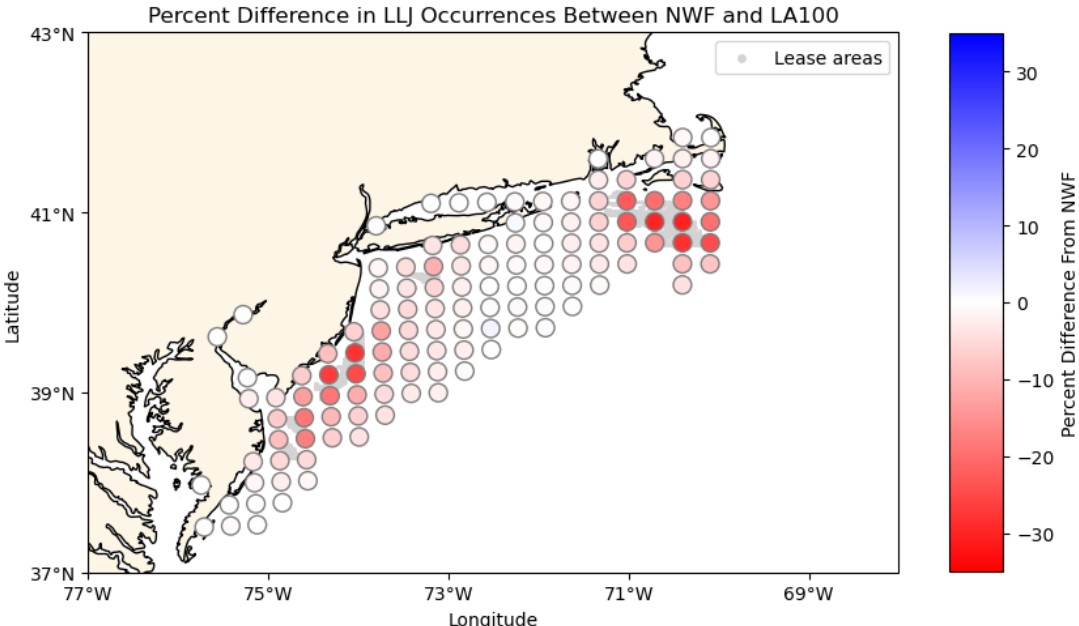

**Figure 9.** The percent reduction in LLJ occurrences between the LA100 and NWF simulations at each point on the map for 1 year. The lease areas are shaded in gray in the background.

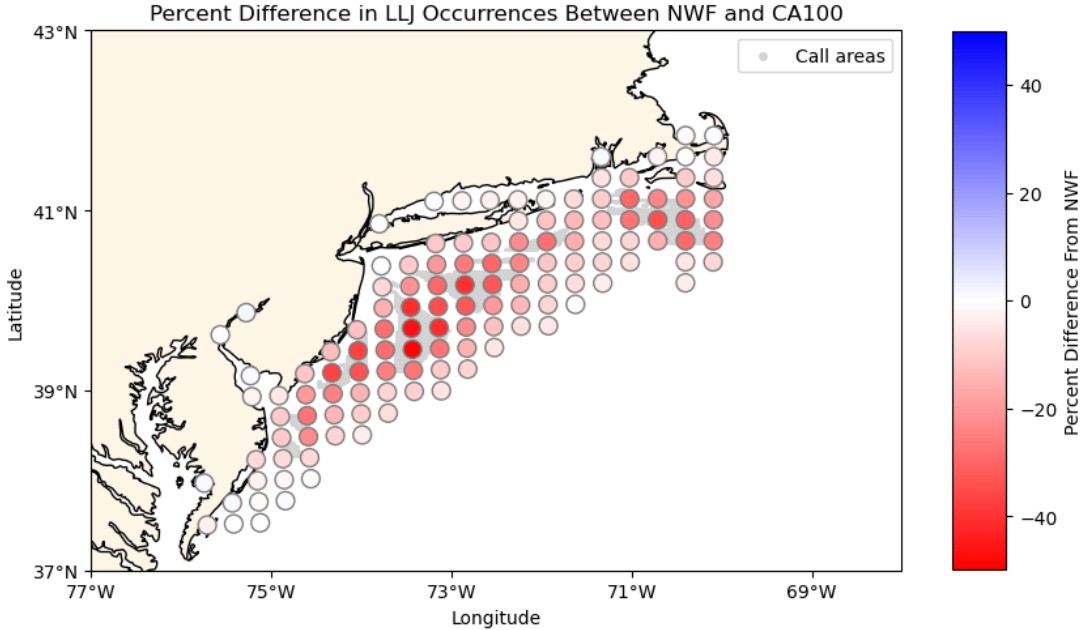

**Figure 10.** The percent reduction in LLJ occurrences between the CA100 and NWF simulations at each point on the map for the July–November period. The lease and call areas are shaded in gray.

## 4.2 Relationship between LLJs and atmospheric stability

In the NWF simulations, LLJs occur during both stable and unstable conditions, but they occur more frequently during stable or neutral stratification (Table 6). Stable conditions can form offshore when warm air from the land flows over cooler air over the ocean. At all five sites, stable conditions occur between 40 % and 47 % of the time, but between 60 % and 66 % of LLJs occur in stable conditions. At ONEcent, the atmosphere was considered stable 46 % of the time, but 65.5 % of LLJs occurred during stable conditions. Neutral conditions occur 5.3 % of the time without LLJs, and 6.8 % with LLJs (Table 6). The other four locations have similar distributions of stable versus unstable conditions for all times of the year and for times with LLJs.

NWF and WF simulations show similar distributions in stability when LLJs occur. In both cases, very stable stratification ($0 \text{ m} < L \leq 200 \text{ m}$) is more common during LLJ events than for all times of the year. Similarly, very unstable conditions ($-200 \text{ m} \leq L < 0 \text{ m}$) are less common during LLJs than for all times (Fig. 11). During neutral conditions ($|L| > 1000 \text{ m}$), differences between LLJs and normal conditions are small. At both the ONEcent and the southLA sites, the NWF LLJs occur slightly more often for Obukhov lengths of 0–100 m than for the WF LLJs (Fig. 11).




**Table 6.** Stability classification for each location for all times of the year (columns 1–3) and for times with an LLJ in the NWF simulation (columns 4–6) at each location (rows 1–5).

|  | All times Stable | All times Unstable | All times Neutral | NWF LLJs Stable | NWF LLJs Unstable | NWF LLJs Neutral |
|---|---|---|---|---|---|---|
| **ONEcent** | 46.3 % | 48.4 % | 5.3 % | 65.5 % | 27.7 % | 6.8 % |
| **SouthLA** | 46.8 % | 47.0 % | 6.2 % | 61.5 % | 28.2 % | 10.3 % |
| SWcorner | 43.8 % | 50.3 % | 5.9 % | 62.6 % | 27.7 % | 9.7 % |
| NEbuoy | 40.6 % | 53.4 % | 6.1 % | 60.6 % | 28.6 % | 10.9 % |
| SWbuoy | 41.4 % | 52.2 % | 6.4 % | 62.2 % | 28.8 % | 9.0 % |

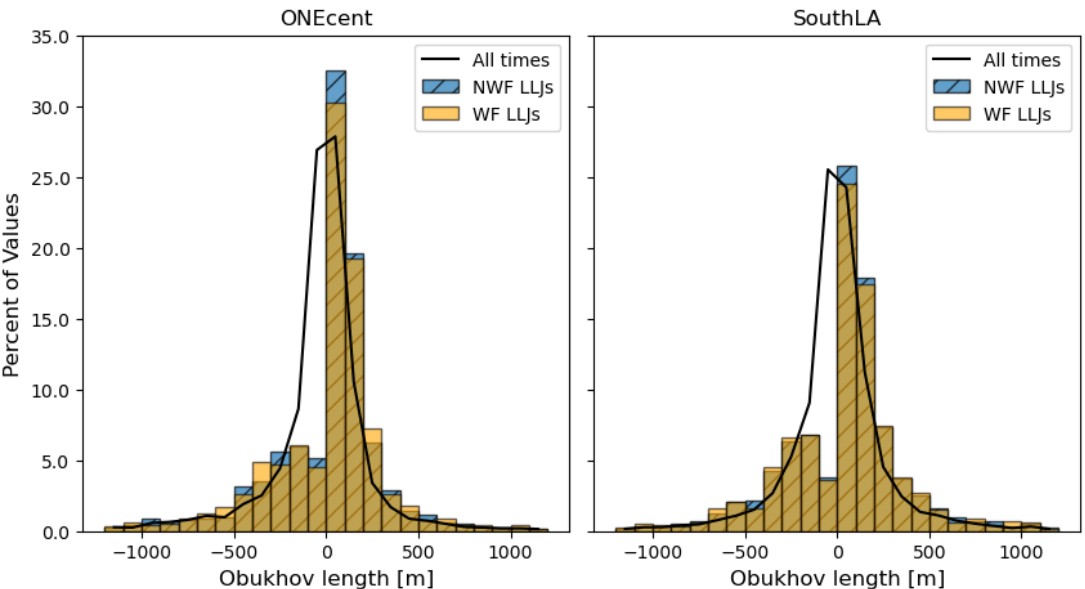

**Figure 11.** Distributions of Obukhov lengths at the ONE centroid (left) and the southern lease area (right). Distribution of Obukhov lengths for the entire year are marked by the black line. Obukhov lengths for LLJs in the NWF simulation and WF simulation are plotted in blue with black hatches and orange, respectively. The percentage of values in each bin are represented by the y-axis.

## 4.3 Wind speed variability of LLJ wind speed maxima

By our definition, LLJs have a minimum wind speed of $10 \text{ m s}^{-1}$. LLJ maxima wind speeds in this study ranged from $10 \text{ m s}^{-1}$ to almost $46 \text{ m s}^{-1}$, but most wind speeds were below $30 \text{ m s}^{-1}$. The LLJ occurrences in the NWF and WF simulations differ the most at slower wind speeds but are nearly identical for wind speeds faster than $20 \text{ m s}^{-1}$ (Fig. 12). While wind speeds of



10–14 m s$^{-1}$ occur around 25 % of the time at hub height (138 m) at each location, these wind speeds only make up around 21 % of the LLJs.

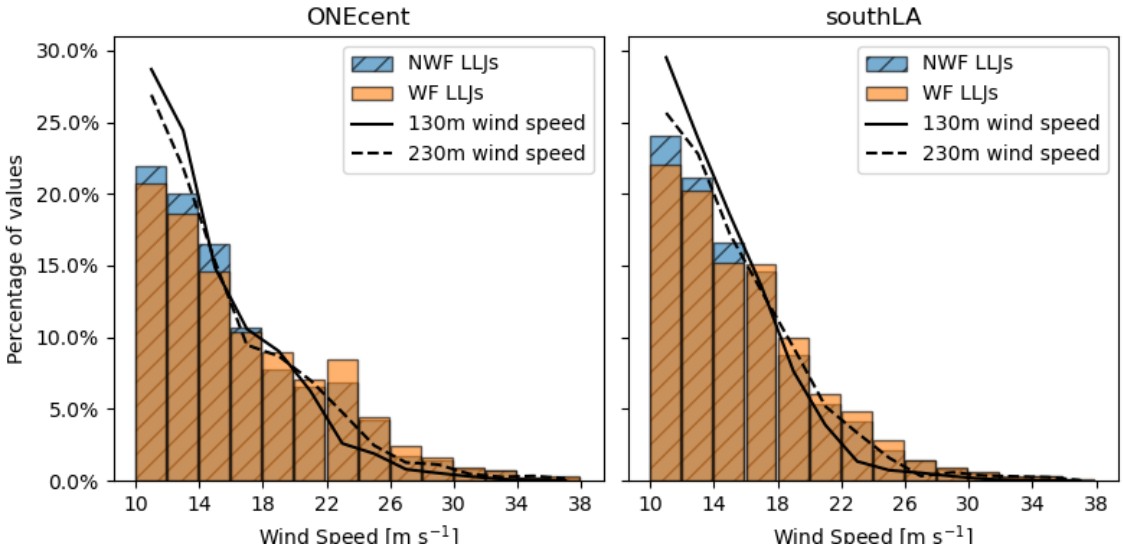

**Figure 12.** Distribution of LLJ nose wind speeds for NWF (blue with black hatches) and WF (solid orange) simulations at ONEcent (left) and southLA (right). Overall distribution of wind speeds faster than 10 m s$^{-1}$ at 130 m and 230 m are shown in solid and dashed black lines, respectively.

### 4.4 Nose heights

Nose heights range from 45 m to 735 m. Mean nose heights in the NWF simulation are around 300 m at both locations, but mean jet height is slightly lower at the southLA site. In the NWF simulation, mean nose height increases with jet classification

at both locations (i.e. faster jets have higher noses). The southLA site also has a larger range of heights for the middle 50 % of data (Fig. 13).

When wind farms are present, the nose heights rise higher in the atmosphere as LLJs are eroded by the wind turbines (rotors in these simulations extend from 30 m to 245 m). In the wind farm simulation, nose heights are all close to 450 m across all classifications, although the southLA site tends to have lower heights than the ONEcent site.

The nose height differences between the NWF and WF simulations are statistically significant. We used a two-sample T-test, with 8273 values in the NWF sample (combined number of LLJ events among the five sites in the NWF simulation) and 6378 values in the WF sample (combined number of LLJ events among the five sites from the WF simulations). The mean of the NWF sample was 348.9 m, and the mean of the WF sample was 439.8 m. Both samples had similar variances, and approximately normal distributions, so the t-test was appropriate. For a t-test with a null hypothesis that the two means

are equal, we found a p-value much lower than the threshold value of 0.05, so we can confidently reject the null hypothesis (that there is no significant difference between the nose heights of the NWF and WF LLJs). When analyzing each location





independently, the null hypothesis can also be rejected for each location with p-values much smaller than $1^{-04}$. The wind farms erode the bottom of the LLJs, pushing the nose heights higher.

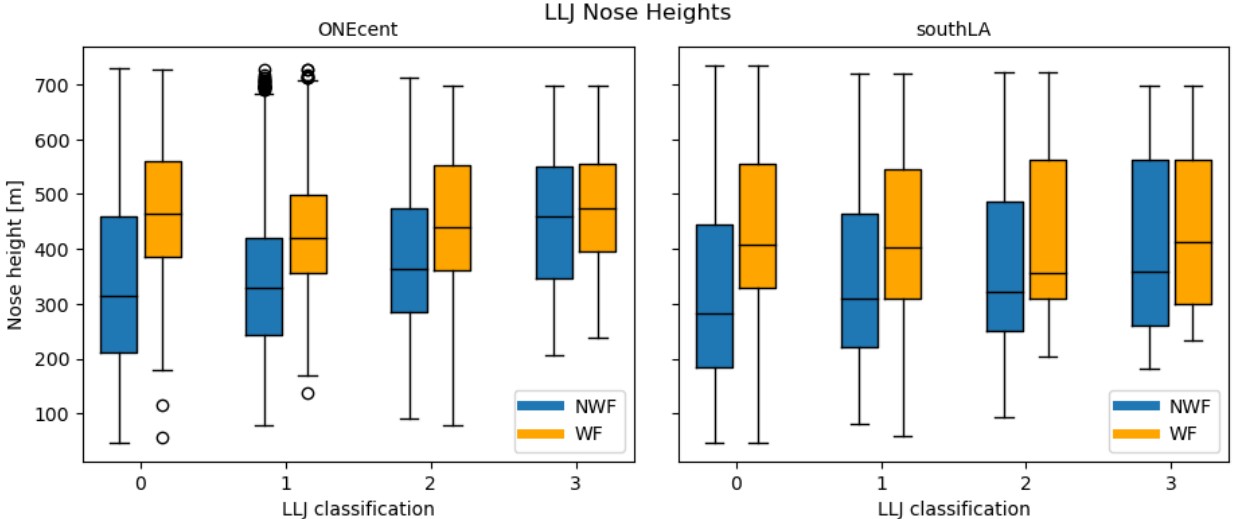

**Figure 13.** Nose heights by LLJ classification for NWF (blue), and WF (orange) at the ONE centroid (left) and southern lease area centroid (right).

We find similar results for the array of LLJ points across the region (Figs. 14, 15, 16). Mean LLJ nose heights range from 328 m to 474 m, but most locations have a mean close to 360 m. The largest increase in mean nose heights occur at locations within a wind farm. Outside of wind farms, nose heights are similar between the NWF and LA100 or CA100 simulations. Points in the vicinity of the wind farms see up to a 129 m difference in the LA100 simulation and up to 205 m in the CA100 simulation. In general, there is a larger effect in the call area simulation, likely due to the greater number of turbines.




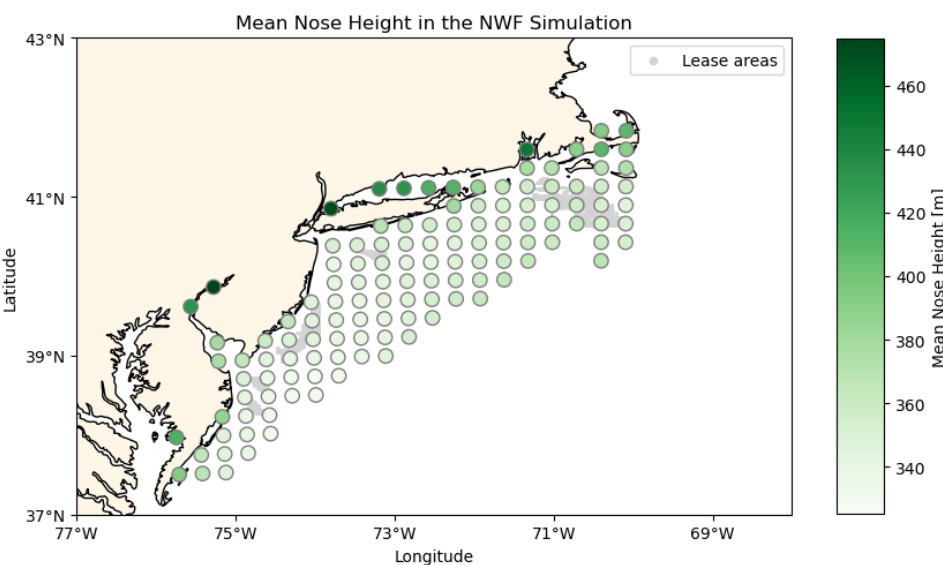

**Figure 14.** The mean nose height in the NWF simulation is shaded in green. The lease areas are shaded in gray in the background.

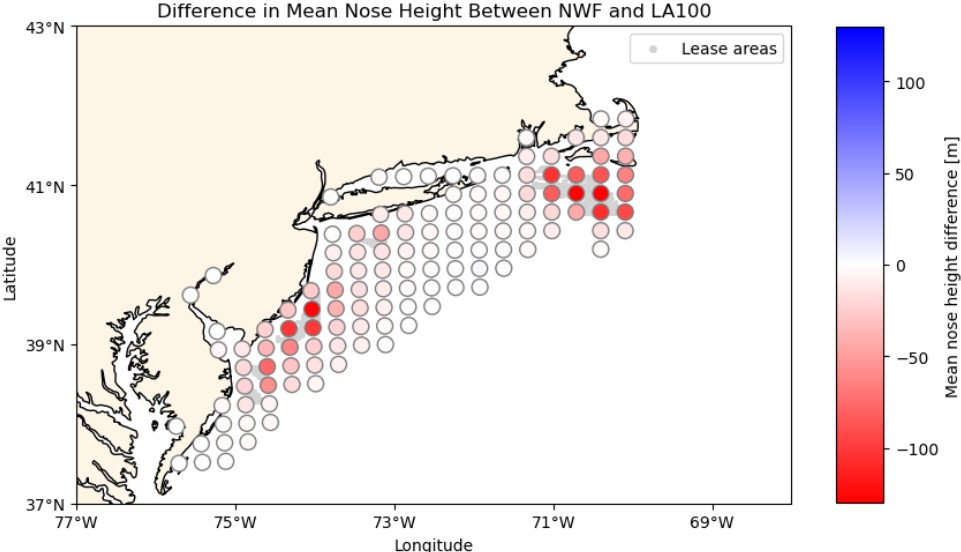

**Figure 15.** The difference in mean LLJ nose height between the LA100 and NWF simulations (NWF − LA100) at each point on the map for one year. The lease areas are shaded in gray in the background.

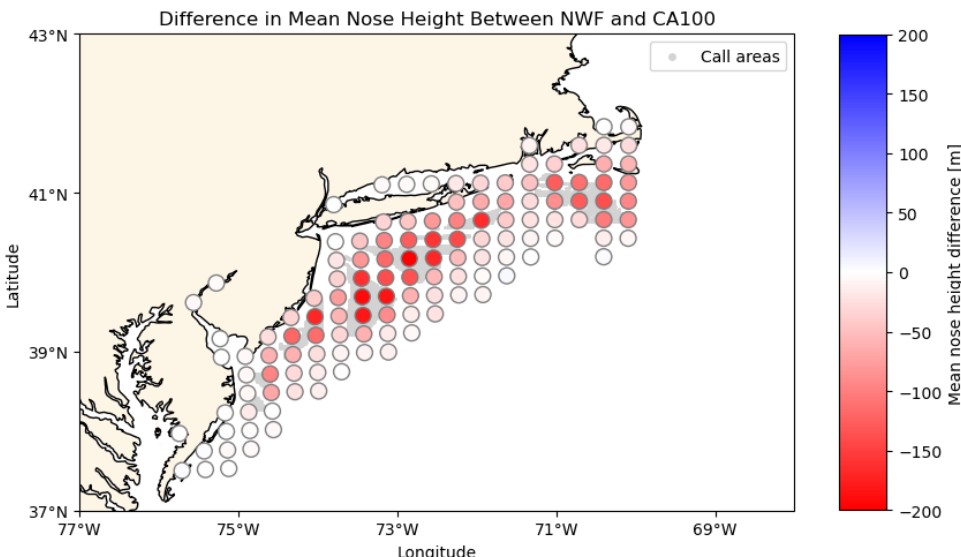

**Figure 16.** The difference in mean LLJ nose height between the CA100 and NWF simulations (NWF − CA100) at each point on the map for the July–November period. The call areas are shaded in gray in the background.

### 4.5 Wind direction variability of LLJ wind speed maxima

Overall wind directions tend to be southwesterly and aligned with the coast, as are the LLJs (Fig. 17). This mode wind direction for LLJs resembles the dominant wind direction as seen in the wind roses of Fig. 1 and the dominant wind direction of stably stratified conditions (not shown). Other investigations of observations of LLJs in this region (Colle and Novak, 2010; Debnath et al., 2021; Aird et al., 2022; de Jong et al., 2023), have also documented a strong preference for southwesterly flow.



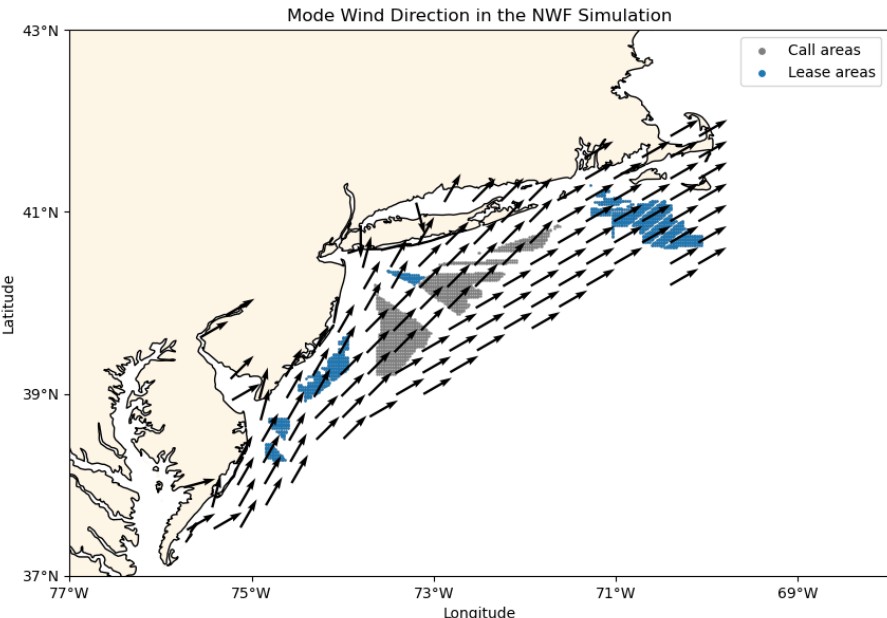

**Figure 17.** Map of mode LLJ nose wind direction at each location using a 15 degree bin size. Arrows point away from where the wind originates.

## 4.6 Temporal variability of LLJs

### 4.6.1 Seasonality

LLJs occur in all months of the year in all locations (Fig. 18). LLJs are more frequent during the spring and summer months, with peak occurrence in May. The difference between the NWF and WF simulations is also the largest during May. Both locations have a minimum in LLJ occurrences from November to February and local maxima in March and October. vLLJs follow a similar seasonal cycle, with most vLLJs occurring in the spring and summer months. In the wind farm simulation (ONE100 for ONEcent and LA100 for southLA), vLLJs are significantly eroded. At the ONEcent location, vLLJs are almost completely eroded across all months of the year. At the southLA site, vLLJs only occur in May and June when wind farms are present: the wind farms erode the vLLJs in all other months.




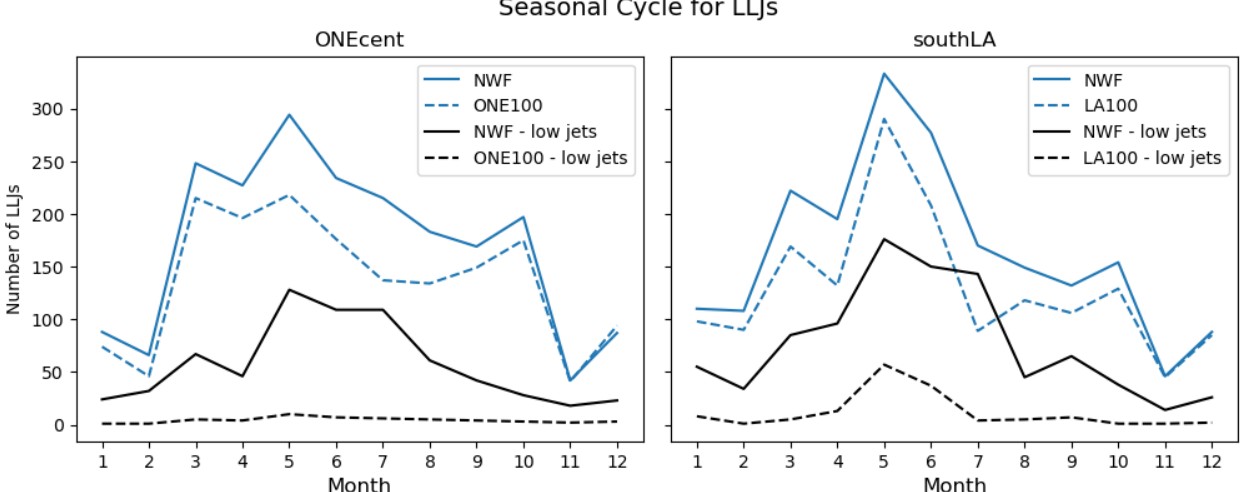

**Figure 18.** The number of LLJs in each month of the year is shown for the ONEcent (left), and southLA (right) locations. The NWF results are marked by a solid blue line, and the WF results are marked by a dashed blue line. vLLJs, with nose heights 260 m or below are in black, where the solid line is for the NWF simulation and the dashed line is for the wind farm simulation.

### 4.6.2 Wind direction seasonality

LLJ wind directions show seasonal variability. Mode wind direction varies depending on the bin size, so we consider results
for bins of 1, 5, and 10 degrees (Fig. 19). Southwesterly and northeasterly winds are most common for LLJs, but this trend varies by month and location. Southwesterly LLJs are most common in the summer months (June–September), as well as in January and March at both locations. In the month of May, LLJs varied across both locations, but were generally northerly or easterly, depending on bin size. October LLJs were consistently northeasterly regardless of bin size or location. December, February, and April LLJs varied by location and bin size.



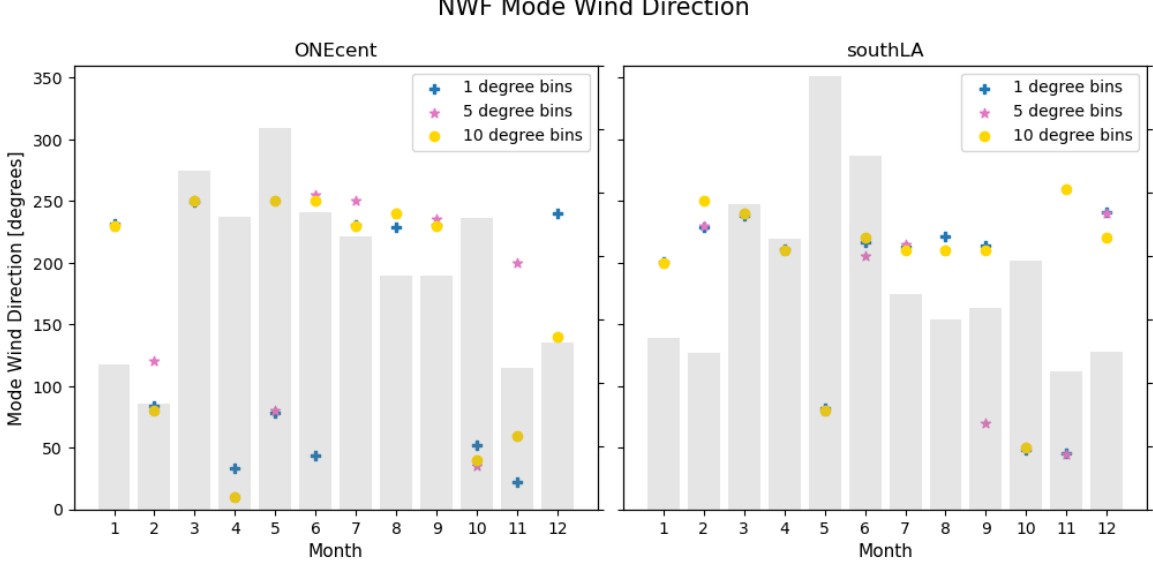

**Figure 19.** Mode nose wind direction in each month for the NWF simulation at each location for bin sizes of 1 (blue plus), 5 (pink star), and 10 (yellow circle) degrees. A histogram of the number of values is in gray. The left y-axis refers to the wind direction, and the right y-axis refers to the histogram.

### 4.6.3 Daily cycle

LLJs can occur at any time of day at both locations but are most common between the hours of 1800 and 0200 local time (Fig. 20). The fewest jets occur in the morning, from 0700 to 1200 local time. In general, LLJs are twice as likely to form at night than during the day. LLJs at the southLA location persist longer into the early morning and disappear for a shorter amount of time later in the day. vLLJs follow a similar diurnal cycle as all jets but are degraded significantly when wind farms are present. vLLJs also have a less pronounced diurnal cycle than for all jets (Fig. 20). These results are consistent with the analysis of Debnath et al. (2021), who find most LLJs also occur at night with a peak of 2200 local time and the fewest events from 0600 to 1200 local time.



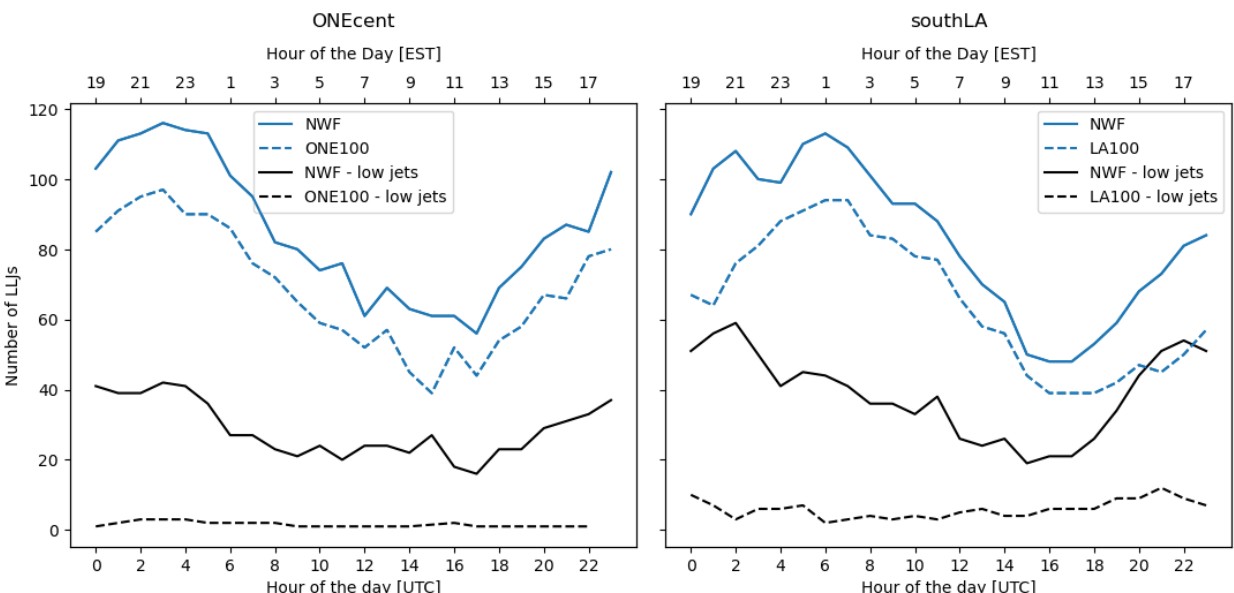

**Figure 20.** The number of LLJs in each hour of the day is shown for ONEcent (left), and southLA (right). The bottom x-axis time is in UTC, and the top x-axis is in local time (EST). The NWF results are marked by the solid blue line and the wind farm results are in the dashed blue line. LLJs with nose heights 260 m or below are in black, where the solid line is for the no wind farm simulation and the dashed line is for the wind farm simulation.

### 4.6.4 Wind veer seasonality

Wind veer associated with LLJs is strong in the spring and summer, reaching 15 degrees across the rotor disk region (30–245 m) but then decreases in the fall and early winter (Fig. 21). While both locations have peaks in the summer, wind veer at the southLA location is consistently stronger from February to May. Wind veer in the rotor region (30–245 m) is reduced at both locations when wind farms are present. The largest difference in wind veer occurs from March to October, with the largest reduction in July. Wind farms induce little difference in veer from November to January (Fig. 21).

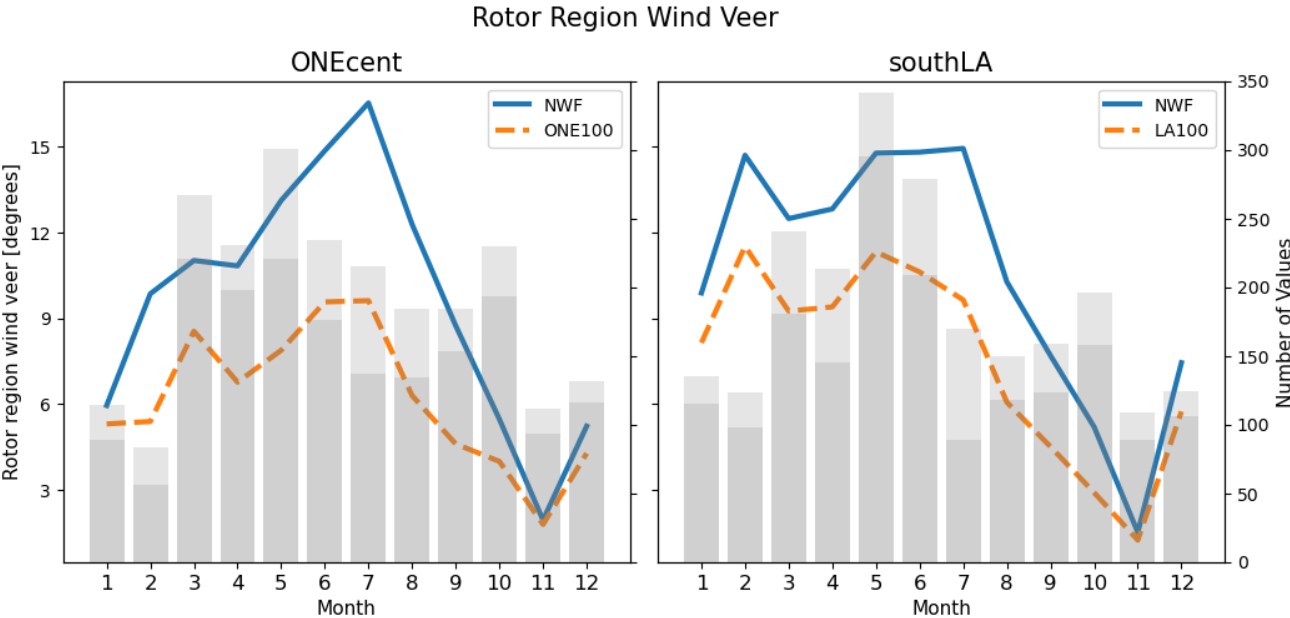

**Figure 21.** Rotor region (30 m–245 m) mean veer by month at ONEcent (left) and the southLA (right). NWF simulations are in solid blue, and the WF simulations are in dashed orange. The number of data points for each month corresponds to the right y-axis and is shown in gray.

## 4.7 Wind Shear seasonality

Mean wind shear in the rotor region (30–245 m) ranges from  0.025 s$^{-1}$ to  0.04 s$^{-1}$ at both locations. Shear values are generally constant throughout the year, but minima occur in July at both locations. At the ONEcent site, wind shear increases in November and December (Fig. 22).

The NWF and WF simulations are similar throughout the year at both locations, with the exception of November and December at the southLA location, where the two simulations diverge. The wind shear is positive throughout the year at both

the ONEcent and southLA locations.

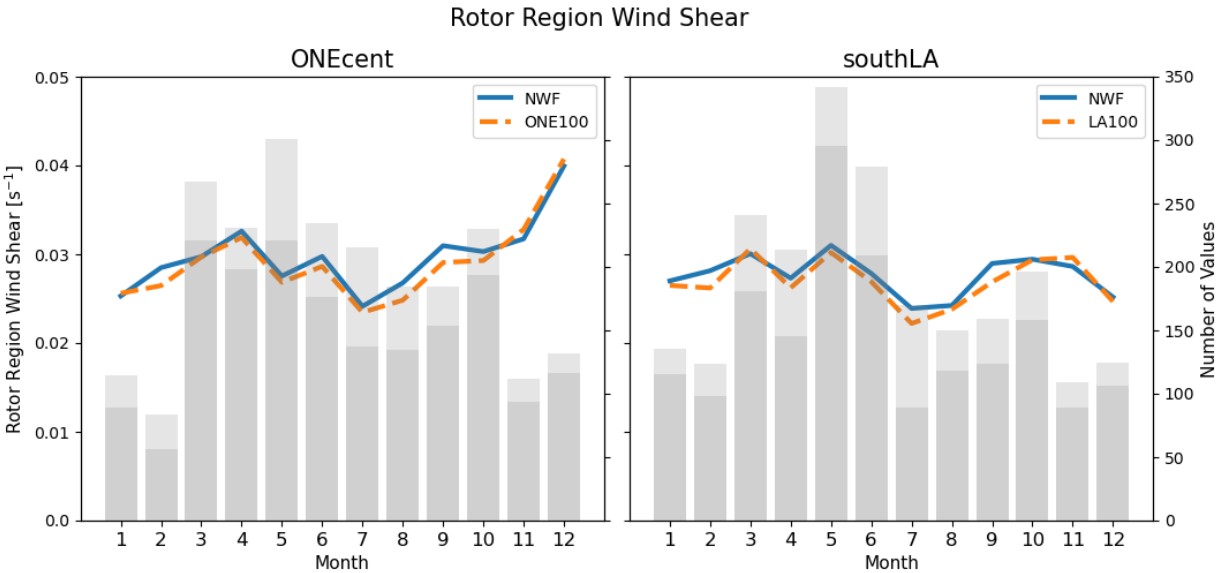

**Figure 22.** Rotor region mean shear by month at ONEcent (left) and the southLA (right). NWF simulations are in solid blue, and the WF simulations are in dashed orange. The number of data points for each month corresponds to the right y-axis and is shown in gray.

## 5 Extreme LLJ case study

On 7 February 2020 at 1900 UTC, an extreme LLJ event was observed at the ONEcent and SWcorner locations. The event was slightly stronger at the SWcorner site, with a LLJ nose wind speed of 45.8 m s$^{-1}$ and a nose height of 564 m. Despite this impressive wind speed, the LLJ was classified as a level 0 event due to the low wind shear above the LLJ nose of 3.7 m s$^{-1}$. While wind speeds faster than 16 m s$^{-1}$ were observed from 1300 UTC on 7 February to 0800 UTC on 8 February, only three of these 20 hours counted as LLJ events due to the weak above-nose shear (Fig. 23).

The LLJ was associated with an anomalous wintertime severe weather outbreak that impacted the mid-Atlantic. At around 0900 UTC on 7 February 2020 a low pressure system over Maryland began to rapidly intensify. By inspection of NOAA/NWS surface analyses akin to Fig. 24, we can see that by 1200 UTC, the low pressure center reached 980 mb, and began to move toward the northeast. At 1500, the low pressure system was centered over New Jersey with a central pressure of 977 mb. At 230 m, wind speeds at the SWcorner location had reached 26 m s$^{-1}$. By 1800 – one hour before the maximum wind speeds were observed at the SWcorner – the low pressure system was centered over Massachusetts, forcing strong southwesterly winds offshore (Fig. 24). The system continued moving along the coast toward Nova Scotia, eventually reaching 968 mb, while winds farther south began to die down. While this LLJ was clearly associated with a specific synoptic event, rather than the typical inertial oscillation/frictional decoupling mechanism, it did generate very fast wind speeds that would affect the wind resource areas, such that turbines would have passed cut-out wind speeds and experience strong wind shear.



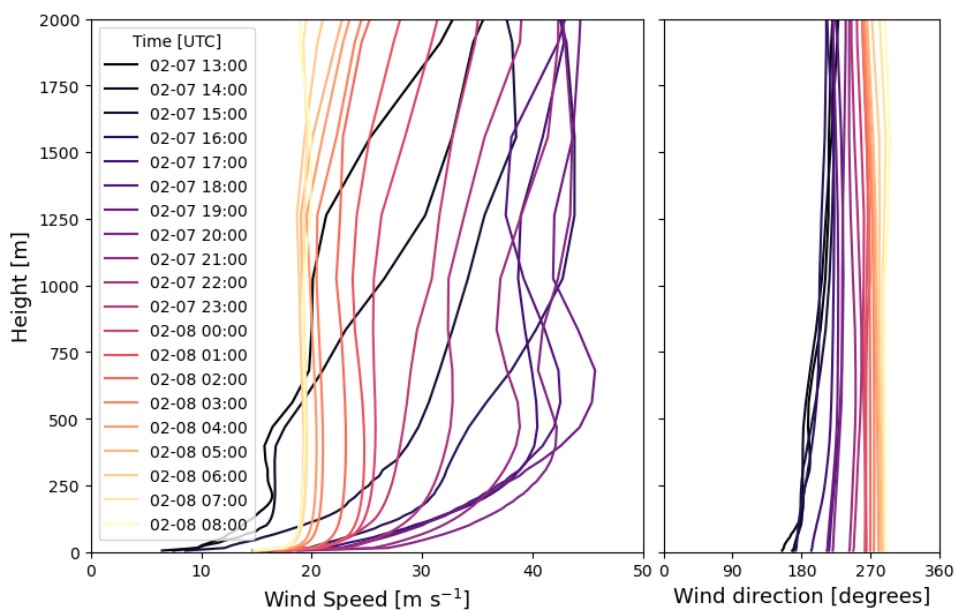

**Figure 23.** Wind profiles from 7 February 2020 13:00 UTC to 8 February 2020 08:00 UTC at the ONEcent location. The wind speed profiles are shown on the left panel, and the wind direction profiles are on the right panel.

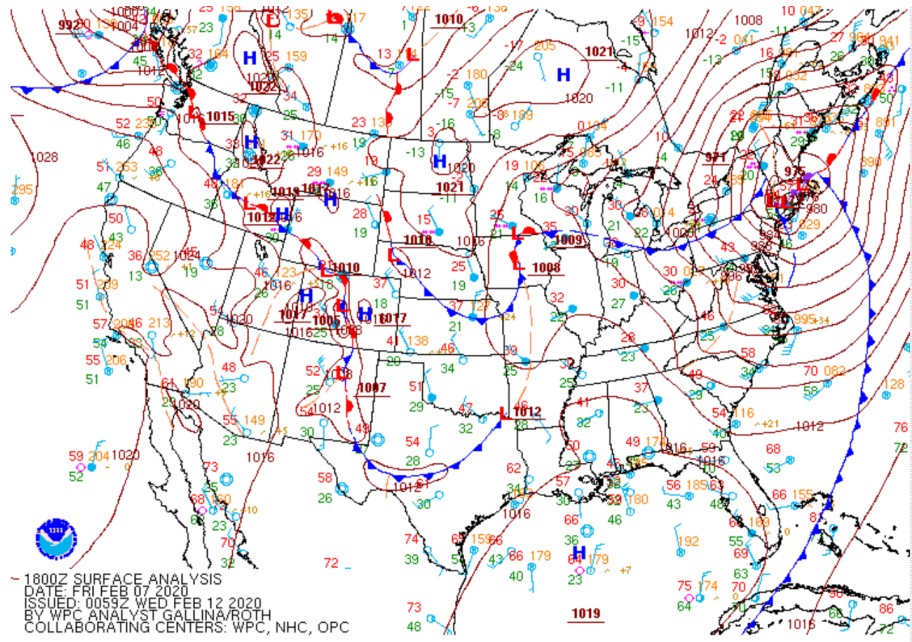

**Figure 24.** Surface analysis from 7 February 2020 at 1800 UTC, 1 hour before the maximum wind speeds were observed at the SWcorner location. Figure courtesy of NOAA/National Weather Service, used with permission according to https://www.weather.gov/disclaimer .



## 6 Conclusions

In this study, we assess occurrences of LLJs in the US East Coast wind resources areas and how these LLJs are influenced by
the presence of wind farms. We identify LLJs for a one-year period at five different locations using WRF simulations with and
without wind farms incorporated into the model. LLJs primarily occur at night, in the spring and summer months, in stably
stratified conditions, and when a southwesterly wind is blowing. LLJs occur less frequently when wind farms are present, and
the jets that do occur tend to have the heights of their wind speed maxima pushed to higher altitudes in the presence of wind
farms, as seen in the case study of Larsén and Fischereit (2021) and the large-eddy simulations of Abkar et al. (2016); Sharma
et al. (2017) but shown to be systematically the case here. We also document how very low-level jets – LLJs with jet nose
heights below 260 m – are significantly eroded by wind farms. Consequently, we find that LLJ nose heights are generally
higher in the wind farm simulations. Mean wind veer in the rotor region is reduced with wind farms, while rotor region wind
shear is generally unaffected.

Because we use model simulation output, this analysis is not constrained to the lowest 200 m as studies based on profiling
lidars are. While Aird et al. (2022) and Debnath et al. (2021) find lower average LLJ nose heights than we find here, our results
are consistent with the scanning lidar observations of Pichugina et al. (2017) that could measure winds to higher altitudes. These
results, then, underscore the importance of using instruments that can probe to higher altitudes to understand the momentum
available for wake replenishment by LLJs.

The seasonal LLJ climatology presented here agrees with other studies on LLJs in this region. Aird et al. (2022), Colle and
Novak (2010), Debnath et al. (2021), and Zhang et al. (2006) all find a peak in LLJs during the warm season and a minimum in
the winter months. There is less of a consensus on the diurnal cycle for LLJs, but multiple studies, both onshore and offshore,
suggest that LLJs occur more often in the early evening and at night (Zhang et al., 2006; Colle and Novak, 2010; Debnath
et al., 2021). Southwesterly winds are the primary wind direction for LLJs in this region , but westerly, northwesterly, and
northeasterly LLJs can also occur (Aird et al., 2022; Debnath et al., 2021; Zhang et al., 2006; Colle and Novak, 2010). LLJ
studies in this region vary in terms of heights considered, but many find jets at heights relevant to wind energy. However, LLJs
with noses above the rotor plane can still impact turbines due to positive wind shear below the nose. In this study, we find that
vLLJs are significantly eroded by wind farms, but occasionally occur at locations with fewer turbines upwind (southLA and
SWcorner (see appendix)).

These findings improve our understanding of the expected energy production from offshore wind projects. While LLJs do
provide significant wind resource, they also can increase loads on turbines (Kelley et al., 2006; Gutierrez et al., 2016). LLJs
also impact the meteorology of the area, which influences energy demand on the East Coast. Because LLJs, and especially
vLLJs, are eroded by wind farms, we can expect LLJs to exert a smaller micrometeorological influence in the vicinity of wind
farms.

The five locations focused on here are selected for their proximity to offshore wind development areas. Despite their geo-
graphic dispersal, they may not fully represent the diverse conditions along the entire U.S. East Coast. Future research could
consider a broader array of locations to better understand the spatial variability of LLJs in the region. Additionally, this study





uses one year of simulations (selected due to the availability of lidar observations for validation of the NWF simulations). Therefore, interannual variability cannot be considered here, so future studies could assess longer data sets to simulate a wider range of conditions.

*Code and data availability.* The data set used can be obtained from https://dx.doi.org/10.25984/1821404. The code used can be found at
https://github.com/doqhne/LLJ_research.

## Appendix A:  Additional sites on land

We also analyzed locations on land at Martha's Vineyard and on Long Island, where we might expect the ONE and Call Area wind farms to impact LLJs. We found that the no wind farm and wind farm simulations were very similar in terms of LLJ occurrence, so these locations are not included in the paper. Distance from the wind farm may play a role in these results.

## Appendix B:  Stability

The SWcorner, NEbuoy, and SWbuoy locations all display a distribution of stability similar to the ONEcent and the southLA locations, with most LLJs occurring during stable conditions (Fig. B1) This figure can be compared to Fig. 11.

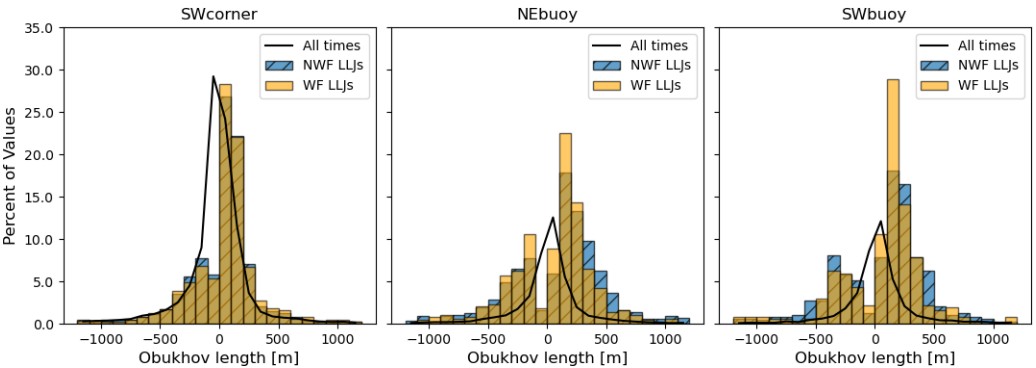

**Figure B1.** Obukhov lengths at the SWcorner, NEbuoy, and SWbuoy locations.

## Appendix C:  Wind speed

The SWcorner site shows a wind speed pattern similar to the ONEcent and southLA locations (see Fig. 12), while LLJs at the
buoys have wind speeds that are more similar to background wind speeds (Fig. C1).





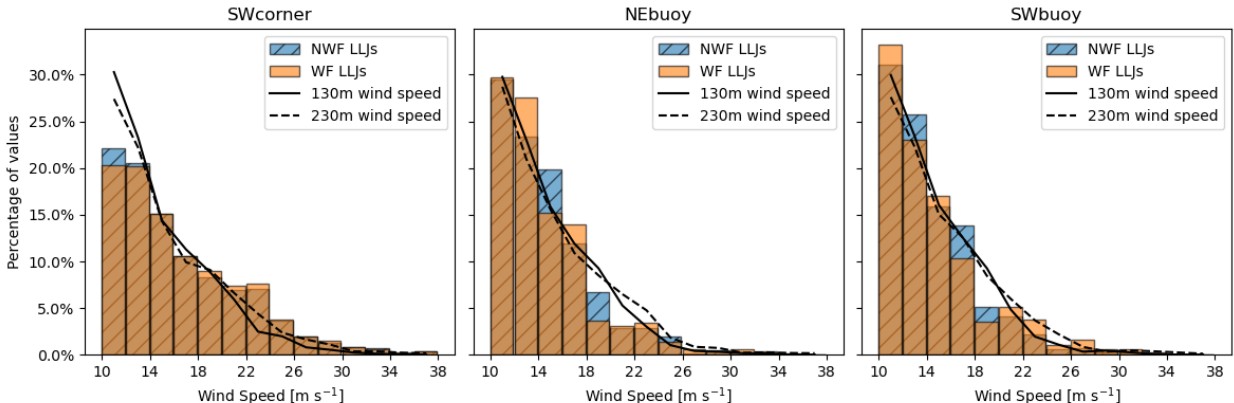

**Figure C1.** Wind speeds at the SWcorner, NEbuoy, and SWbuoy locations.

## Appendix D: Seasonality

The SW corner exhibits a similar seasonal cycle to the ONE and southLA locations (see Fig. 18), where LLJs peak in May (Fig. D1). Note that vLLJs are not eroded as much here. This resistance to erosion may be because there are fewer turbines upwind of the SWcorner when a SW wind is blowing. At the buoys, with many turbines upwind, vLLJs are completely eroded.

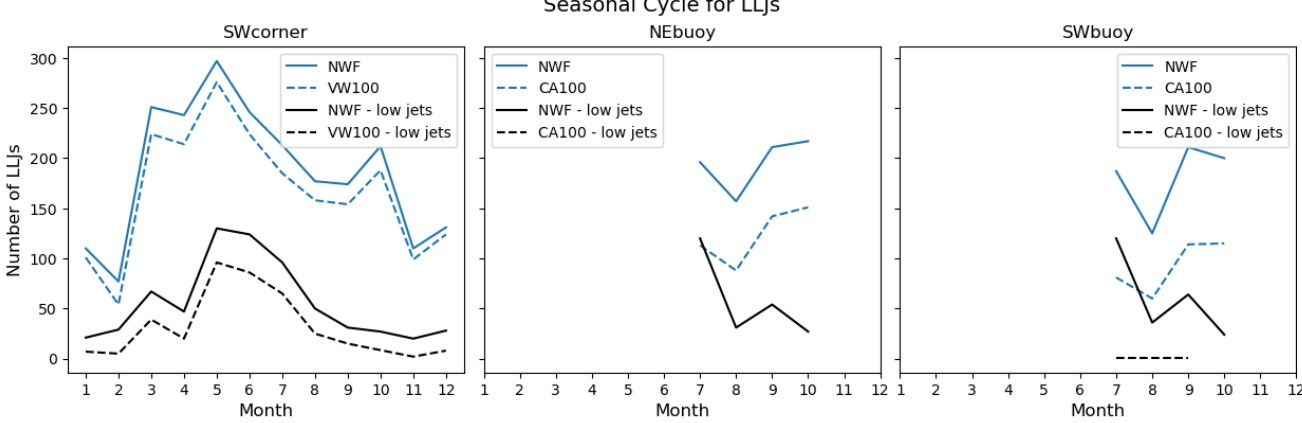

**Figure D1.** Seasonal cycle at the SWcorner, NEbuoy, and SWbuoy locations.

## Appendix E: Daily cycle

The SWcorner, NEbuoy, and SWbuoy locations all show similar diurnal cycles to the ONEcent and southLA locations (Fig. E1). Compare this figure to Fig. 20.

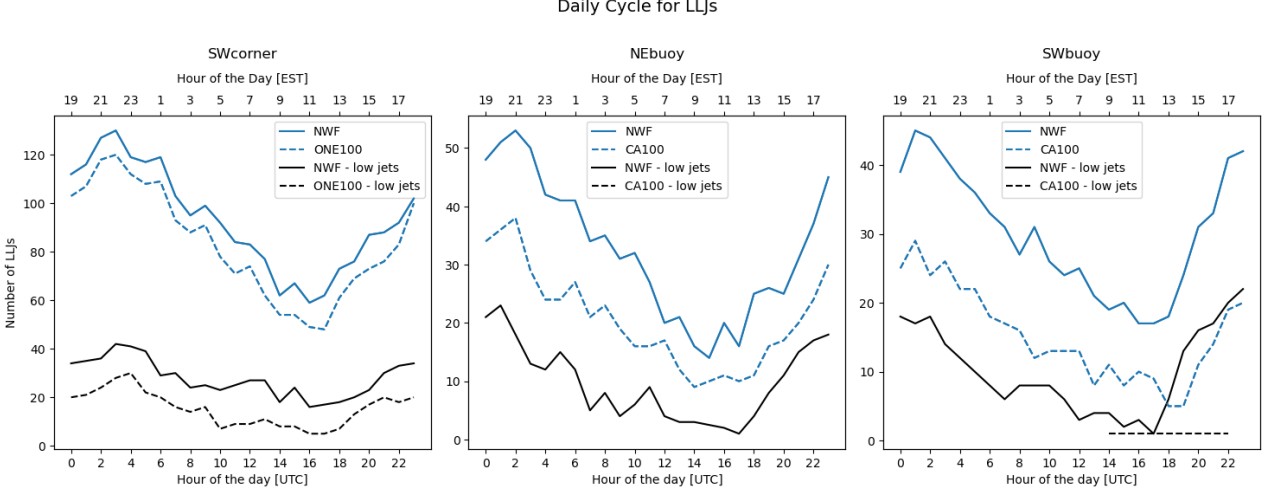

**Figure E1.** Diurnal cycle at the SWcorner, NEbuoy, and SWbuoy locations.

## Appendix F: Veer

Similar to the ONEcent and southLA locations (see Fig. 21), mean rotor region wind veer is reduced when wind farms are

present at the buoys. At the SW corner, rotor region veer is similar for all months of the year (Fig. F1). Fewer turbines upwind of this location may play a role in this difference.

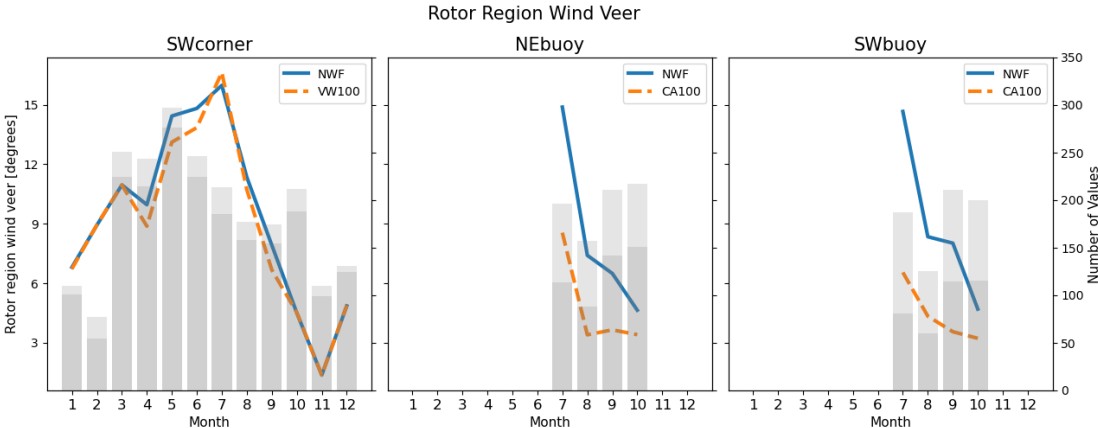

**Figure F1.** Nose heights at the other three locations.





## Appendix G: Shear

Similar to the ONEcent and southLA locations (see Fig. 22), mean rotor region wind shear is reduced when wind farms are present at the buoys. At the SWcorner, rotor region shear is similar for all months of the year (Fig. G1). We suspect that fewer turbines upwind of this location may play a role in this difference.

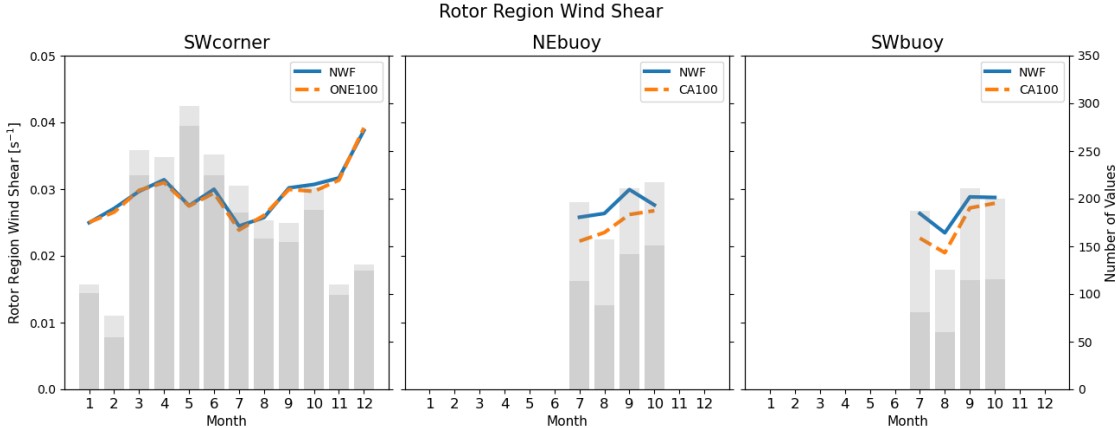

**Figure G1.** Nose heights at the SWcorner, NEbuoy, and SWbuoy locations.

## Appendix H: Nose Heights

Mean nose heights are higher in the wind farm simulations at the SWcorner, NEbuoy, and SWbuoy locations. These results are in agreement with results from the ONE and southLA sites (Fig. 13). Nose height differences are much larger at the buoys, with very little overlap between the middle 50 percent of data for each classification (Fig. H1).

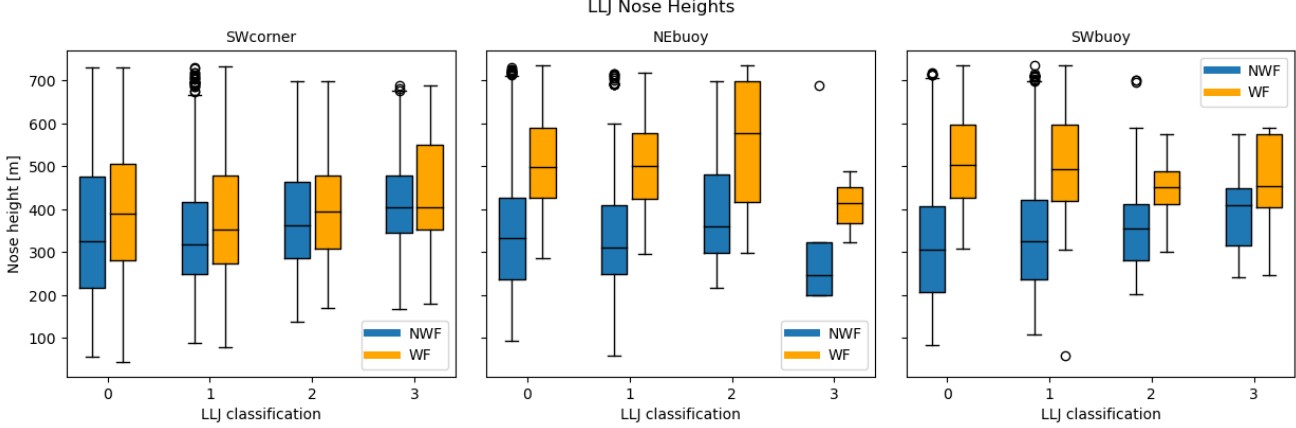

**Figure H1.** Nose heights at the other three locations.





*Author contributions.* JKL conceptualized the project and acquired funding and resources for the project. DR completed the WRF simulations and provided the data set. DQ carried out the formal analysis and investigation, including developing software and carrying out the visualization, with supervision from JKL and DR. DQ and JKL prepared the initial draft. JKL, DR, and DQ reviewed and edited the publication.

*Competing interests.* At least one of the (co-)authors is a member of the editorial board of Wind Energy Science. Otherwise, the authors declare they have no competing interests.

This work was supported by an agreement with NREL under APUP UGA-0-41026-125. This work was authored in part by the National Renewable Energy Laboratory, operated by Alliance for Sustainable Energy, LLC, for the U.S. Department of Energy (DOE) under Contract No. DE-AC36-08GO28308. Funding was provided by the U.S. Department of Energy Office of Energy Efficiency and Renewable Energy Wind Energy Technologies Office and by the National Offshore Wind Research and Development Consortium under agreement no. CRD-19-16351. The views expressed in the article do not necessarily represent the views of the DOE or the U.S. Government. The U.S. Government and the publisher, by accepting the article for publication, acknowledge that the U.S. Government retains a nonexclusive, paid-up, irrevocable, worldwide license to publish or reproduce the published form of this work, or allow others to do so, for U.S. Government purposes. Neither NYSERDA nor OceanTech Services/DNV have reviewed the information contained herein and the opinions in this report do not necessarily reflect those of any of these parties. A portion of computation used the Blanca condo computing resource at the University of Colorado Boulder. Blanca is jointly funded by computing users and the University of Colorado Boulder. A portion of computation used the Summit supercomputer, which is supported by the National Science Foundation (awards ACI-1532235 and ACI-1532236), the University of Colorado Boulder, and Colorado State University. The Summit supercomputer is a joint effort of the University of Colorado Boulder and Colorado State University. A portion of this research was performed using computational resources sponsored by the DOE's Office of Energy Efficiency and Renewable Energy and located at NREL.



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
