# Peer review of "Simulations suggest offshore wind farms modify low-level jets"

_Wind Energy Science, 2024_

## Author Comment (AC1)

**REVIEWER 1**

*Reviewer's comments appear in italics,* **our responses appear in boldface blue text and new text included in the manuscript appears in** ***boldface italicized blue text***.

*Quint et al. have produced a nice comparison of low-level jet occurrences and features based on simulations with and without offshore wind farms present. This study is of value to the wind energy community, especially in light of the extensive activity occurring offshore of the United States northeast region. The topic is deeply delved into by the authors, and the accompanying graphics are of high quality. I especially appreciated that the authors performed what validations they could, while acknowledging the limitations imposed by the vertical extent of available observations.*

**We thank the reviewer for their time and thoughtful consideration in reviewing our manuscript. We especially appreciate their kind words.**

*General comments*

*I have some concerns about the strength of the wording throughout the manuscript as it pertains to making bold physical statements based on simulations. Even the title would imply a well-documented observation-based study of offshore wind farms modifying LLJs instead of a comparison of simulations that lack extensive validation. There is still a lot of value in such a study, I just think care needs to be taken with the messaging.*

**Thank you for the thoughtful comment. We have changed the title to "Simulations suggest Offshore wind farms modify low-level jets" and have softened the language throughout to remind readers that we are interrogating a simulation dataset. Some examples, with** underlining **denoting the new softened text:**

- **Abstract: "**In the absence of observations or significant wind farm construction as yet, **we compare one year of simulations from the Weather Research and Forecasting model with and without wind farms incorporated,"**
- **Abstract: "In the NOW-23** simulation **dataset"**
- **Conclusions: "In this** simulation-based **study, we assess occurrences of LLJs in the US East Coast wind resources areas and how these LLJs are influenced by the presence of wind farms** as they appear in numerical weather prediction simulations.**"**
- **Conclusions: "**Modeled **LLJs occur less frequently when wind farms are present** in the simulations**"**
- **Conclusions: "We also document how very low-level jets – LLJs with jet nose heights below 260 m – are significantly eroded by wind farms** in the simulations.**"**

*There are extensive details that the reader must keep track of with not a lot of helpful reminders along the way. The phrase "the WF (wind farm) simulation" is used throughout the text, and it is easy to forget that you're actually talking about three WF simulations being used to analyze five unique locations. And those five unique locations are reduced to two, with the other three placed in the appendices, but then the Conclusions section again discusses all five. In the Results section, Figure 7 presents five locations and is immediately followed by Table 5, which presents three locations, while most of the text in Section 4 discusses "both locations." I recommend streamlining by picking the locations of greatest research value*

*to you to focus the bulk of the paper on, and then briefly point to the additional details available in the appendices.*

**Thank you for pointing out the potential confusion here. We have been specifically requested to assess these five locations and so streamlining by eliminating locations is not an option. To try to clarify, we have added a column to Table 1 to specify clearly which simulation is used, and we added a sentence to the end of section 2.2 to state this:**

***"Throughout the manuscript, we refer to wind farm (WF) simulations, and the appropriate simulation depends on the location as specified in the rightmost column of Table 1."***

**We have also clarified that Table 5 only includes three sites because only three sites have a full year of data available:**

***"The NEbuoy and SWbuoy sites do not appear in Table 5 because only four months of data are available for those two sites."***

**Finally, there are several locations in Section 4 in which we focus only on two sites because they represent bounding conditions in which there is not much variability between the five sites. We have added sentences in those areas pointing out that the omissions are generally because of redundancy and directing readers interested in those specific locations to the appendix and highlighting cases in which the buoys behave differently. For example:**

***"Similarly, very unstable conditions (−200 m ≤ L < 0 m) are less common during LLJs than for all times (Fig. 11), and this pattern holds for the other three sites (Fig. B1). During neutral conditions (|L|>500 m), differences between LLJs and normal conditions are small, although few LLJs occur during neutral conditions."***

**and**

***"The LLJ occurrences in the NWF and WF simulations differ the most at slower wind speeds but are nearly identical for wind speeds faster than 20 m s-1 (Fig. 12 and Fig. C1).***

**and**

***"Mean nose heights in the NWF simulation are around 300 m at all five locations (Fig. 13 and Fig. H1), but mean jet height is slightly lower at the southLA site than at the ONEcent site. In the NWF simulation, mean nose height increases with jet classification at all locations (i.e. faster jets have higher noses) except for the NE buoy. The southLA site also has a larger range of heights for the middle 50 % of data (Fig. 13) Nose height differences between WF and NWF simulations are much larger at the buoys, with very little overlap between the middle 50 percent of data for each classification (Fig. H1), likely due to the very large wind farms in their vicinity."***

**Several similar changes have been made throughout Section 4 to more completely include all five sites, including references to the figures in the Appendix.**

*Many interesting findings are presented, but, in some cases, little to no physical suggestion or speculation accompanies the results. For example, the seasonal trends in wind veer reduction from the*

*presence of wind farms presented in Section 4.6.4 are quite pronounced. Can you provide some comments in the text as to why those seasonal trends might exist in the simulations?*

**Yes, we should have included a discussion of this pattern, which is related to the seasonal variation in atmospheric stability. Summer months have more frequent occurrences of stable conditions, and stable conditions tend to have more wind veer due to frictional decoupling. We have added text to Sect. 4.6.4:**

***"This pattern is related to the seasonal variability in atmospheric stability. Summer months have more frequent stable conditions, and stable conditions are associated with more veer (Lundquist, 2020), especially offshore (Bodini et al., 2019, 2020)."***

*Check the intermingling of past versus present tense throughout the text. Lines 119-121 provide an example.*

**Thank you, we have thoroughly reviewed the tenses and changed to present tense throughout.**

*Specific comments*

*Line 10: It would be of interest to the reader to put some quantification here (percentage) on how many fewer LLJs occur in the wind farm simulations versus the no-wind-farm simulations.*

**We have modified the abstract to read**

***"In the NOW-WAKES simulation dataset, we find offshore LLJs in this region occur about 25 % of the time, most frequently at night…Wind farms erode LLJs, as LLJs occur less frequently (19-20 % of hours) in the wind farm simulations than in the no-wind-farm (NWF) simulation (25 % of hours)."***

*Line 19: Citing BOEM's website here would provide the reader with knowledge of the status of the lease areas at the time they read the manuscript.*

**We have added a sentence to include BOEM's website as requested*: "Wind turbines will be grouped into clusters within the 27 active wind farm lease areas that span the mid-Atlantic Outer Continental Shelf; current status of lease areas can be found at https://www.boem.gov/renewable-energy/lease-and-grant-information.***

*Line 100: The Rosencrans paper does a nice job of discussing the validation, but it would be helpful here to reiterate at least some of the validating findings, particularly those that pertain to bias, in order to set expectations.*

**Thank you for the helpful suggestion, we expanded this sentence to include the bias: "These simulations without wind farms have been validated in comparison to floating lidar observations at two locations in the domain (Rosencrans et al., 2023), with a slow bias of less than 0.5 m s-1."**

*Line 102: "The period from 1 September 2019 00:00 UTC to 31 August 2020 23:50 UTC provides a temporal resolution of 10 minutes; we used hourly time steps for our analysis." Why?*

***In our experience, LLJs vary more slowly, over the course of several hours, and so we thought that using the higher-temporal resolution data would be redundant and not add much insight (and would require more computational resources to process the data).***

*Section 3.1: It would be helpful here to remind readers that there are no consistently agreed upon numerical thresholds to define LLJs. Please consider explaining why you felt the selected LLJ definitions were the best ones for this analysis as opposed to the numerous other options in the literature.*

**Thank you for this suggestion. We have expanded this paragraph (new text is underlined) to include the lineage of our approach as well as direct readers to other options.**

**"To identify LLJs in the simulations described above, we follow the established methodology described by Vanderwende et al. (2015), based on the foundational LLJ analysis of Bonner (1968) and Whiteman et al. (1997). This approach was also used by Song et al. (2005). However, we do include slight modifications for the offshore environment. LLJs are detected if the maximum wind speed occurs in the lowest 750 m of the atmosphere and is at least 10 m s-1. The wind speed reduction above this wind speed maximum (the "nose" of the jet (Banta et al., 2002)) must be at least 3 m s-1; we considered heights up to 2 km for our analysis. Given the difference in mechanisms offshore and onshore (smaller force of friction leading to weaker super-geostrophic acceleration), we use a smaller shear threshold than in Vanderwende et al. (2015). Several other approaches for identifying LLJs have appeared in the literature, often designed specifically for the features of the instrument platform used (i.e. Nunalee and Basu (2014) use radar wind profiler observations that offer deep layers of observations but lack measurements in the lowest 100 m). Many of the recent developments designed for relatively shallow profiling lidar observational datasets rather than deeper observational datasets (from scanning lidar, radar wind profilers, or radiosondes) or modeling sets used here. No consistent formulation currently exists, but several of the shallow approaches are summarized in Sheridan et al. (2024), including those of Kalverla et al. (2019) and Hallgren et al. (2020)."**

*Line 179: The authors should add references to other recent studies that classify modelled LLJs into hits, correct rejections, misses, and false alarms, namely, Hallgren et al. (2020), Kalverla et al. (2020), and Sheridan et al. (2024).*

**Thank you for this suggestion. We have included the following sentence:**

**"This approach was also used by Kalverla et al. (2019),Hallgren et al. (2020), and Sheridan et al. (2024)."**

*Line 204: Can the authors provide any speculations as to why one location had more LLJs and another had fewer?*

**Because the differences in the numbers of LLJs are very small (i.e. 26% vs 25.1%) we do not find these differences significant enough to provide material for speculation.**

*Line 224: "Neutral conditions…" Suggest reformatting this sentence to follow the flow of the previous two for improved clarity. "Neutral conditions occur X%, and Y% of LLJs occurred during neutral conditions."*

**Thank you for the suggestion, we have changed the sentence to "Neutral conditions occur 5.3 % of the time, but 6.8 % of the LLJs occur during neutral conditions."**

*Figure 12: Cropped at the lower extent*

**Thank you for noticing that; we fixed it.**

*Line 298: This sentence is quite confusing. I think you mean that the trends are similar between ONEcent and southLA throughout the year, except for November and December. But the use of "with the exception of November and December at the southLA location, where the two simulations diverge" could also imply that the NWF and WF simulations are diverging from each other during these two months at this single location. Figure 22 indicates that it is the former assumption, not the latter, but this is another example where clarity is essential.*

**Thank you for identifying this confusing phrasing. We have rewritten** *"Mean wind shear in the rotor region (30–245 m) ranges from 0.025 $s_{-1}$ to 0.04 $s_{-1}$ at both locations. Shear values are generally constant throughout the year, but minima occur in July at both locations. At the ONEcent site, wind shear increases in November and December (Fig. 22). The NWF and WF simulations are similar throughout the year at both locations, with the exception of November and December at the southLA location, where the two simulations diverge. The wind shear is positive throughout the year at both the ONEcent and southLA locations. "*

**to be**

*"Mean wind shear in the rotor region (30–245 m) ranges from 0.025 $s_{-1}$ to 0.04 $s_{-1}$ at both the ONEcent and southLA locations. Shear values are generally constant throughout the year, but minima occur in July at both locations (Fig. 22). The primary difference between the two locations occurs in November and December, when shear decreases at southLA but increases at ONEcent. This regional variability could be due to the fact that the southern southLA location enters a more unstably stratified regime (with less wind shear) sooner in the winter than the more northern ONEcent location (with more stable conditions and more wind shear). At both locations, few differences occur between the NWF and WF shear. "*

*Line 302: "An extreme LLJ event was observed…" was it observed? Or did it appear in your simulations? If observed, please provide references to the data sources. If simulated, please rephrase the wording in Section 5 to indicate as such.*

**Yes, strong winds were observed by the lidars west of the Vineyard Wind area with similar wind speeds and wind directions as simulated, although this deep LLJ (with a nose well above 250 m) could not be fully observed by the lidar platforms. We have included a figure from the lidar observations to address this question and rephrased the discussion.**

*Line 359: Can you include the distances from the Martha's Vineyard and Long Island sites from the wind farms alongside the distances between the sites you did analyze and the wind farms, for comparison value?*

**The sites that we analyzed are within the planned wind farms, so a distance of 0 km. The south coast of Martha's Vineyard site is ~ 30 km from the closest part of the ONE wind farm. The closest distance of Long Island to Empire Wind I is ~ 22 km, per https://www.nyserda.ny.gov/-/media/Project/Nyserda/Files/Publications/Fact-Sheets/LSR-offshore-wind-visibility-fact-sheet.pdf. We have added such text to the Appendix:**

*"Distance from the wind farm may play a role in these results: the south coast of Martha's Vineyard site is ~ 30 km from the closest part of the ONE wind farm. The closest distance of Long Island to Empire Wind I is ~ 22 km, per https://www.nyserda.ny.gov/-/media/Project/Nyserda/Files/Publications/Fact-Sheets/LSR-offshore-wind-visibility-fact-sheet.pdf."*

*References:*

*Hallgren, C., Arnqvist, J., Ivanell, S., Körnich, H., Vakkari, V., and Sahlée, E.: Looking for an Offshore Low-Level Jet Champion among Recent Reanalyses: A Tight Race over the Baltic Sea, Energies, 13, 3670, https://doi.org/10.3390/en13143670, 2020.*

*Kalverla, P. C., Holtslag, A. A. M., Ronda, R. J., and Steeneveld, G.-J.: Quality of wind characteristics in recent wind atlases over the North Sea, Q. J. Roy. Meteorol. Soc., 146, 1498–1515, https://doi.org/10.1002/qj.3748, 2020.*

*Sheridan, L. M., Krishnamurthy, R., Gustafson Jr., W. I., Liu, Y., Gaudet, B. J., Bodini, N., Newsom, R. K., and Pekour, M.: Offshore low-level jet observations and model representation using lidar buoy data off the California coast, Wind Energ. Sci., 9, 741–758, https://doi.org/10.5194/wes-9-741-2024, 2024.*

---

## Author Comment (AC2)

**REVIEWER 2**

*Reviewer's comments appear in italics*, **our responses appear in boldface blue text and new text included in the manuscript appears in *boldface italicized blue text*.**

*The study examines the impact of offshore wind farms on low-level jets (LLJs) along the east coast of the United States using Weather Research and Forecasting model simulations. The results indicate that wind farms reduce the frequency of LLJs and increase their height, with significant implications for regional meteorology.*

*The study is well-structured and written, and the results are clearly presented.*

**We thank the reviewer for their time and thoughtful consideration in reviewing our manuscript and their kind words, greatly appreciated.**

*My only concern is the lack of proper LLJ validation of the WRF by observation, as according to the WRF results 'LLJ nose heights' range from 328 m to 474 m and are well above the lidar range. This is addressed by the authors and cannot be resolved with the existing data set. For this reason, more care should be taken with the wording, and it should be made clear that the results are based on simulations, not real observations. For example, the title, passages in the abstract or the summary should be changed and phrased more appropriately.*

**Thank you for the thoughtful comment. We have changed the title to "Simulations suggest Offshore wind farms modify low-level jets" and have softened the language throughout to remind readers that we are interrogating a simulation dataset. Some examples, with underlining denoting the new softened text:**

- **Abstract: "In the absence of observations or significant wind farm construction as yet, we compare one year of simulations from the Weather Research and Forecasting model with and without wind farms incorporated,"**
- **Abstract: "In the NOW-23 simulation dataset"**
- **Conclusions: "In this simulation-based study, we assess occurrences of LLJs in the US East Coast wind resources areas and how these LLJs are influenced by the presence of wind farms as they appear in numerical weather prediction simulations."**
- **Conclusions: "Modeled LLJs occur less frequently when wind farms are present in the simulations"**
- **Conclusions: "We also document how very low-level jets – LLJs with jet nose heights below 260 m – are significantly eroded by wind farms in the simulations."**

*I have only minor comments, and if these are considered, the study can be a very valuable contribution to the offshore wind energy community.*

*Specific comments:*

*Abstract: The abstract of the submitted manuscript does not correspond to the abstract of the revised one. Please clarify.*

**We will ensure that the abstract in the Copernicus system corresponds to the abstract of the revised text when we upload a revised manuscript.**

*L.55 What is the Long-Ez aircraft ? Please explained*

**As Mahrt et al. (2014) explain, "The Long-EZ is a light pusher aircraft with the engine mounted on the rear of the airplane. It has the large main wing set back farther than that of conventional aircraft. The small, low-drag airframe and rear-mounted pusher engine reduce the influence of flow distortion, engine vibration, and engine exhaust for instruments that are mounted on the nose." In the atmospheric science community, this aircraft is somewhat infamous as a year after these observations, during another Long-EZ flight as part of the CBLAST experiment, the pilot and director of NOAA's Air Resources Laboratory Field Research Division died (https://www.whoi.edu/science/AOPE/CBLAST/TimCrawford.html, https://www.arl.noaa.gov/news-pubs/news-archive/arl-news-death-of-tim-crawford-august-7-2002/) although it was later determined Dr. Crawford's death was due to a stroke and not due to aircraft issues. The senior author realizes that she should not have assumed that the broader readership of WES would be aware of the Long-EZ.**

**We have modified the text to provide some basic information about the Long-EZ:**

**"Over two days south of Martha's Vineyard, Mahrt et al. (2014) observe low-level wind maxima associated with developing stable stratification; the altitude of the wind speed maximum is higher with stronger stability. (These observations were collected with the Long-EZ aircraft, a light pusher aircraft with a rear-mounted engine and a small, low-drag airframe.)"**

*L. 74 Here you could add also a study about the stability by. Platis et al 2021:*

*Platis, A., Hundhausen, M., Lampert, A., Emeis, S., & Bange, J. (2021). The role of atmospheric stability and turbulence in offshore wind-farm wakes in the German bight. Boundary-Layer Meteorology, 1-29.*

**Thank you, we have added this reference.**

*Fig. 4 Why do you see these line structures in the scatter plot? Does the 'Calculated' Data have a higher variability compared to RMOL of WRF ?*

**The "line structures" are artifacts given that the WRF values of RMOL do not vary continuously but are binned to regular intervals for values outside of |300| m-1 due to the iterative calculation method given in https://github.com/wrf-model/WRF/blob/master/phys/module_sf_mynn.F, function zolrib (near line 1984 as of July 2024). The calculated data does have higher variability as it is not forced to specific values or bins.**

*Sect. 3.4 What is the accuracy of the lidar measurements? Is it well above the 1m/s that you consider to be the LLJ criteria?*

**The measurement uncertainty for the lidar measurements is given as 3.3% of the wind speed, well under the 1 m s-1 threshold that we require for shear above the low-level jet nose. This uncertainty is given in DNV's assessment report "HUDSON CENTRAL AND HUDSON SOUTH LEASE AREAS OFFSHORE WIND FARM Energy Assessment Report, Document number 10124962-HOU-R-01", downloadable from the NYSERDA**

datasite https://oswbuoysny.resourcepanorama.dnv.com/download/f67d14ad-07ab-4652-16d2-08d71f257da1.

*L 169 How long did a LLJ Event last? How did you consider this in the data analysis, also regarding Fig. 4 to calculate the percentage of LLJ ? How is the no. of LLJ related to the possible of 8784 hours (Fig.7) ? Please clarify.*

**As of yet, we have not considered duration of LLJ events and instead treat each time's wind profile (and surface stability assessment in Figure 4) separately. We have added a clarifying sentence**

***"We treated each time profile separately and did not consider the duration of LLJ events or require continuity of LLJ events."***

**The number of LLJs on the y-axis of Figure 7 is out of a possible 8784 hours, for a general frequency on the order of 25%. We have added a % axis to Figure 7 to clarify this.**

*Table 6: The sum of the NEbuoy stable, unstable and neutral is 100,1 % . Please correct this (probably a rounding error).*

**Thank you for catching this - we corrected both the "All Times" and "NWF Times". We later modified our threshold for neutral conditions to be consistent with other papers coming from this dataset so that all the numbers changed, but we have double-checked to ensure there are no other rounding errors.**

*Sect. 4.2 Have you studied the dependency of stability on the wind direction?*

**This is an interesting question and, yes, we have looked at this in another associated study (Quint, Lundquist, Bodini, Rosencrans, 2024). In that study, figures reproduced here, stable cases almost entirely consist of southwesterly winds.**

[Figure]

**Figure 6.** (a) The frequency of each 22.5-degree wind direction bin for each stability classification. Wind roses for stable, neutral, and unstable conditions are shown in Panels b, c, and d, respectively. In all panels, radial distance from the center refers to the percentage of values in each 22.5 degree bin.

**We have added a sentence to Section 4.2 to acknowledge this variability and reference the other study:**

***"As noted in Quint et al. (2024), stable conditions in this region occur almost always with southwesterly winds (their Figure 6)."***

*Fig. 12 . Why does the overall wind speed around 10 m/s appear to be at hub height 5 % more often than at 230 m ?*

**If the reviewer is asking why the probability distribution in Figure 12 for the "southLA" region suggests that relatively slow winds (10 m s-1) occur more often at 130m than at 230m, we would point out that at lower altitudes, slower winds tend to occur more often. In contrast, for the 18-26 m s-1 bins, these faster winds occur more often at 230m than at 130m so that the area under the 130m curve is the same as the area under the 260m curve.**

*Sect. 4.6.4 Why is the wind veer highest in the summer months? Please give an interpretation and/or discussion.*

**Yes, we should have included a discussion of this pattern, which is related to the seasonal variation in atmospheric stability. Summer months have more frequent occurrences of stable conditions, and stable conditions tend to have more wind veer due to frictional decoupling. We have added text to Sect. 4.6.4:**

***"This pattern is related to the seasonal variability in atmospheric stability. Summer months have more frequent stable conditions, and stable conditions are associated with more veer (Lundquist, 2020), especially offshore (Bodini et al., 2019, 2020)."***

*Sect. 5 What wind speeds did the lidar measure? Do they compare the modelled values?*

**Yes, strong winds were observed by the lidars west of the Vineyard Wind area with similar wind speeds and wind directions as simulated, although this deep LLJ (with a nose well above 250 m) could not be fully observed by the lidar platforms. We have included a figure from the lidar observations to address this question and rephrased the discussion.**

**References**

**Quint, D., Lundquist, J. K., Bodini, N., and Rosencrans, D.: Meteorological Impacts of Offshore Wind Turbines as Simulated in the Weather Research and Forecasting Model, Wind Energ. Sci. Discuss. [preprint], https://doi.org/10.5194/wes-2024-53, in review, 2024.**